# Temporal Test-Time Adaptation with State-Space Models

**Mona Schirmer**                                                                 *m.c.schirmer@uva.nl*
*UvA-Bosch Delta Lab, University of Amsterdam*

**Dan Zhang**                                                                   *dan.zhang2@de.bosch.com*
*Bosch Center for AI*

**Eric Nalisnick**                                                                  *nalisnick@jhu.edu*
*Johns Hopkins University*

**Reviewed on OpenReview:** *https://openreview.net/forum?id=HFETOmUtrV*

## Abstract

Distribution shifts between training and test data are inevitable over the lifecycle of a deployed model, leading to performance decay. Adapting a model on test samples can help mitigate this drop in performance. However, most test-time adaptation methods have focused on synthetic corruption shifts, leaving a variety of distribution shifts underexplored. In this paper, we focus on distribution shifts that evolve gradually over time, which are common in the wild but challenging for existing methods, as we show. To address this, we propose STAD, a Bayesian filtering method that adapts a deployed model to temporal distribution shifts by learning the time-varying dynamics in the last set of hidden features. Without requiring labels, our model infers time-evolving class prototypes that act as a dynamic classification head. Through experiments on real-world temporal distribution shifts, we show that our method excels in handling small batch sizes and label shift.

## 1 Introduction

Predictive models often have an 'expiration date.' Real-world applications tend to exhibit distribution shift, meaning that the data points seen at test time are drawn from a distribution that is different than the training data's. Moreover, the test distribution usually becomes more unlike the training distribution as time goes on. An example of this is with recommendation systems: trends change, new products are released, old products are discontinued, etc. Unless a model is updated, its ability to make accurate predictions will expire, requiring the model to be taken offline and re-trained. Every iteration of this model life-cycle can be expensive and time consuming. Allowing models to remain 'fresh' for as long as possible is thus an open and consequential problem.

*Test-time adaptation* (TTA) (Liang et al., 2024; Yu et al., 2023) has emerged as a powerful paradigm to preserve model performance under a shifting test distribution. TTA performs online adaptation of a model's parameters using only test-time batches of features. By requiring neither access to labels nor source data, TTA algorithms can be employed in resource-constrained environments, whereas related approaches such as domain generalization, domain adaptation and test-time training cannot. Most TTA methods operate by minimizing an entropy objective (Wang et al., 2021) or updating normalization parameters (Schneider et al., 2020; Nado et al., 2020; Niu et al., 2023).

Synthetically corrupted images (e.g. CIFAR-10-C) are by far the most commonly used benchmark for assessing progress on TTA, despite concerns about benchmark diversity (Zhao et al., 2023b). These shifts increase the degree of information loss over time, and well-performing TTA methods must learn to preserve a static underlying signal. In this work, we focus on an omnipresent distribution shift of quite a different nature: *Temporal distribution shifts* encode structural change, not just information loss. Gradual structural change over time is relevant for any deployed model that is operating continuously. While related subfields

like gradual domain adaptation (GDA) (Kumar et al., 2020) and temporal domain generalization (TDG) (Bai et al., 2023) are dedicated to these shifts, they have received little attention in TTA. As we will demonstrate using datasets like the Functional Map of the World (FMoW), which classifies land use over time (e.g., rural to urban development), the setting of *temporal test-time adaptation* (TempTTA) presents significant challenges for existing TTA methods.

To address this gap, we propose *State-space Test-time Adaptation (STAD)*, a method that builds on the power of probabilistic state-space models (SSMs) to represent non-stationary data distributions over time. STAD dynamically adapts a model's final layer to accommodate an evolving test distribution. Specifically, we employ a probabilistic SSM based on Bayesian filtering to model the evolution of the weight vectors in the final layer, where each vector represents a class, as distribution shift occurs. For generating predictions on newly acquired test batches, we use the SSM's posterior cluster means as the new parameters. STAD leverages Bayesian updating and does not rely upon normalization mechanisms. As a consequence, STAD excels in scenarios where many TTA methods collapse (Niu et al., 2023), such as adapting with very few samples and under label shift. Our contributions are the following:

- In Sec. 2, we detail the setting of TempTTA, which aims to cope with shifts that gradually evolve due to variation in the application domain. Despite being ubiquitous in real-world scenarios, these shifts are understudied in the TTA literature and pose significant challenges to established methods, as we demonstrate in Sec. 5.1.

- In Sec. 3, we propose STAD, a novel method for TempTTA. It adapts to temporal distribution shifts by modeling its dynamics in representation space. No previous work has explicitly modeled these dynamics, which we demonstrate is crucial via an ablation study (Sec. 5.3).

- In Sec. 5, we conduct a comprehensive evaluation of STAD and prominent TTA baselines under authentic temporal shifts. Our results show that STAD excels in this setting (Tab. 2), yet is applicable beyond temporal shifts providing performance gains on some reproduction datasets (Tab. 3), synthetic corruptions (Tabs. 4 and 12) and domain adaptation benchmarks (Tabs. 8 and 9).

## 2 Problem Setting

**Data & Model**   We focus on the traditional setting of multi-class classification, where $\mathcal{X} \subseteq \mathbb{R}^D$ denotes the input (feature) space and $\mathcal{Y} \subseteq \{1, \ldots, K\}$ denotes the label space. Let $\mathbf{x}$ and $y$ be random variables and $\mathbb{P}(\mathbf{x}, y) = \mathbb{P}(\mathbf{x}) \mathbb{P}(y|\mathbf{x})$ the unknown source data distribution. We assume $\boldsymbol{x} \in \mathcal{X}$ and $y \in \mathcal{Y}$ are realisations of $\mathbf{x}$ and $y$. The goal of classification is to find a mapping $f_\theta$, with parameters $\theta$, from the input space to the label space $f_\theta : \mathcal{X} \to \mathcal{Y}$. Fitting the classifier $f_\theta$ is usually accomplished by minimizing an appropriate loss function (e.g. log loss). Yet, our method is agnostic to how $f_\theta$ is trained and therefore easy to use with, for instance, a pre-trained model downloaded from the web.

**Temporal Test-Time Adaptation (TempTTA)**   We are interested in adapting a model at test-time to a test distribution that evolves with time. More formally, let $\mathcal{T} = \{1, \ldots, T\}$ be a set of $T$ time indices. At test time, let the data at time $t \in \mathcal{T}$ be sampled from a distribution $\mathbb{Q}_t(\mathbf{x}, y) = \mathbb{Q}_t(\mathbf{x}) \, \mathbb{Q}_t(y|\mathbf{x})$. The test distributions differ from the source distribution, $\mathbb{Q}_t(\mathbf{x}, y) \neq \mathbb{P}(\mathbf{x}, y) \; \forall t > 0$, and are non-stationary, meaning $\mathbb{Q}_t(\mathbf{x}, y) \neq \mathbb{Q}_{t'}(\mathbf{x}, y)$ for $t \neq t'$. Like in standard TTA, we of course do not observe labels at test time, and hence we observe only a batch of features $\mathbf{X}_t = \{\mathbf{x}_{1,t}, \ldots, \mathbf{x}_{N,t}\}$, where $\mathbf{x}_{n,t} \sim \mathbb{Q}_t(\mathbf{x})$ (i.i.d.). Given the $t$-th batch of features $\mathbf{X}_t$, the goal is to adapt $f_\theta$, forming a new set of parameters $\theta_t$ such that $f_{\theta_t}$ has better predictive performance on $\mathbf{X}_t$ than $f_\theta$ would have. Since we can only observe features, we assume that the distribution shift must at least take the form of *covariate shift*: $\mathbb{Q}_t(\mathbf{x}) \neq \mathbb{P}(\mathbf{x}) \; \forall t > 0$. In addition, a *label shift* may occur, which poses an additional challenge: $\mathbb{Q}_t(\mathbf{y}) \neq \mathbb{P}(\mathbf{y}) \; \forall t > 0$. Temporal shifts, as described above, have been the focus of temporal domain generalization (Bai et al., 2023) and gradual domain adaptation (Abnar et al., 2021). However, both paradigms operate during training, whereas TempTTA is applicable at test time. In Tab. 1, we contrast TempTTA with adjacent fields highlighting subfields that address temporal shifts. Notably, TempTTA can be seen as a special case of continual TTA (CTTA) (Wang et al., 2022b) with the important distinction that the domain index $t$ is inherently temporal. This is opposed to a categorical domain index (e.g. different corruption types) as is mostly studied in CTTA.

Table 1: Comparison of TempTTA with related fields

| | Adaptation stage | Available samples | Test distribution non-stationary | time-ordered |
|---|---|---|---|---|
| Domain generalization (DG) | train | $\mathbb{P}(\mathbf{x}, \mathrm{y})$ | ✗ | ✗ |
| Temporal DG | train | $\mathbb{P}_1(\mathbf{x}, \mathrm{y}), \ldots, \mathbb{P}_{T_S}(\mathbf{x}, \mathrm{y})$ | ✓ | ✓ |
| Domain adaptation (DA) | train | $\mathbb{P}(\mathbf{x}, \mathrm{y}), \mathbb{Q}(\mathbf{x})$ | ✗ | ✗ |
| Gradual DA | train | $\mathbb{P}(\mathbf{x}, \mathrm{y}), \mathbb{Q}_1(\mathbf{x}), \ldots, \mathbb{Q}_T(\mathbf{x})$ | ✓ | ✓ |
| Test-time training | train, test | $\mathbb{P}(\mathbf{x}, \mathrm{y}), \mathbb{Q}(\mathbf{x})$ | ✗ | ✗ |
| Test-time adaptation (TTA) | test | $\mathbb{Q}(\mathbf{x})$ | ✗ | ✗ |
| Continual TTA | test | $\mathbb{Q}_1(\mathbf{x}), \ldots, \mathbb{Q}_T(\mathbf{x})$ | ✓ | ✗ |
| **TempTTA** | test | $\mathbb{Q}_1(\mathbf{x}), \ldots, \mathbb{Q}_T(\mathbf{x})$ | ✓ | ✓ |

## 3 Tracking the Dynamics of Temporal Shifts

We now present our method: the core idea is that adaptation to temporal distribution shifts can be done by tracking its gradual change in the model's representations. We employ linear SSMs to capture how test points evolve and drift. The SSM's cluster representations then serve as an adaptive classification head that evolves with the non-stationarity of the distribution shift. Fig. 1 illustrates our method. In Sec. 3.2, we first introduce the general model and then, in Sec. 3.3, we propose an efficient implementation that leverages the von Mises-Fisher distribution to model spherical features.

### 3.1 Adaptation in Representation Space

Following previous work (Iwasawa & Matsuo, 2021; Boudiaf et al., 2022), we adapt only the last layer of the source model. This lightweight approach is reasonable for Temp-TTA since the distribution shifts gradually over time, hence constrained adaptation is needed. From a practical perspective, this circumvents backpropagation through potentially large networks such as foundation models and allows adaptation when only embeddings are provided e.g. by an API. More formally, let the classifier $f_\theta$ be a neural network with $L$ total layers. We will treat the first $L-1$ layers, denoted as $f_\theta^{L-1}$, as a black box that transforms the original feature vector $\mathbf{x}$ into a new (lower-dimensional) representation, which we denote

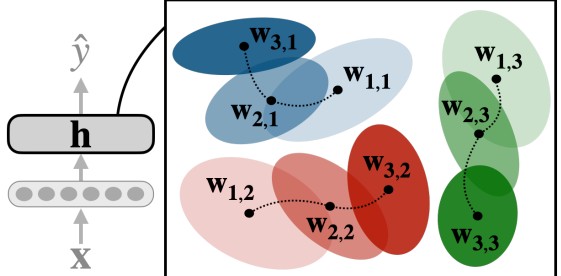

Figure 1: STAD adapts to distribution shifts by inferring dynamic class prototypes $\mathbf{w}_{t,k}$ for each class $k$ (different colors) at each test time point. It operates on the representation space of the penultimate layer.

as $\mathbf{h}$. The original classifier then maps these representations to the classes as: $\mathbb{E}[y|\mathbf{h}] = \text{softmax}_y\left(\mathbf{W}_0^\top \mathbf{h}\right)$, where $\text{softmax}_y(\cdot)$ denotes the dimension of the softmax's output corresponding to the $y$-th label index and $\mathbf{W}_0$ are the last-layer weights. As $\mathbf{W}_0$ will only be valid for representations that are similar to the training data, we will discard these parameters when performing TempTTA, learning new parameters $\mathbf{W}_t$ for the $t$-th time step. These new parameters will be used to generate the adapted predictions through the same link function: $\mathbb{E}[y|\mathbf{h}] = \text{softmax}_y\left(\mathbf{W}_t^\top \mathbf{h}\right)$. In the setting of TempTTA, we observe a batch of features $\mathbf{X}_t$. Passing them through the model yields corresponding representations $\mathbf{H}_t$, and this will be the 'data' used for the probabilistic model we will describe below. Specifically, we will model how the representations change from $\mathbf{H}_t$ to $\mathbf{H}_{t+1}$ next.

### 3.2 A Probabilistic Model of Shift Dynamics

We now describe our general method for a time-evolving adaptive classification head. We assume that, while the representations $\mathbf{H}_t$ are changing gradually over time, they are still maintaining some class structure in the form of clusters. Our model will seek to track this structure as it evolves. For the intuition of the

approach, see Fig. 1. The blue red, and green clusters represent classes of a classification problem. As the distribution shifts from time step $t = 1$ to $t = 3$, the class clusters shift in representation space. Using latent variables $\mathbf{w}_{t,k}$ for the cluster centers, we will assume each representation is drawn conditioned on $K$ latent vectors: $\boldsymbol{h}_{t,n} \sim p(\mathbf{h}_t | \mathbf{w}_{t,1}, \ldots, \mathbf{w}_{t,K})$, where $K$ is equal to the number of classes in the prediction task. After fitting the unsupervised model, the $K$ latent vectors will be stacked to create $\mathbf{W}_t$, the last-layer weights of the adapted predictive model (as introduced in Sec. 3.1). We now move on to a technical description.

**Notation and Variables** Let $\mathbf{H}_t = (\mathbf{h}_{t,1}, \ldots, \mathbf{h}_{t,N_t}) \in \mathbb{R}^{D \times N_t}$ denote the neural representations for $N_t$ data points at test time $t$. Let $\mathbf{W}_t = (\mathbf{w}_{t,1}, \ldots, \mathbf{w}_{t,K}) \in \mathbb{R}^{D \times K}$ denote the $K$ weight vectors at test time $t$. As discussed above, the weight vector $\mathbf{w}_{t,k}$ can be thought of as a latent prototype for class $k$ at time $t$. We denote with $\mathbf{C}_t = (\mathbf{c}_{t,1}, \ldots, \mathbf{c}_{t,N_t}) \in \{0, 1\}^{K \times N_t}$ the $N_t$ one-hot encoded latent class assignment vectors $\mathbf{c}_{t,n} \in \{0, 1\}^K$ at time $t$. The $k$-th position of $\mathbf{c}_{t,n}$ is denoted with $c_{t,n,k}$ and is 1 if $\mathbf{h}_{t,n}$ belongs to class $k$ and 0 otherwise. Like in standard (static) mixture models, the prior of the latent class assignments $p(\mathbf{c}_{t,n})$ is a categorical distribution, $p(\mathbf{c}_{t,n}) = \texttt{Cat}(\boldsymbol{\pi}_t)$ with $\boldsymbol{\pi}_t = (\pi_{t,1}, \ldots, \pi_{t,K}) \in [0, 1]^K$ and $\sum_{k=1}^K \pi_{t,k} = 1$. The mixing coefficient $\pi_{t,k}$ gives the a priori probability of class $k$ at time $t$ and can be interpreted as the class proportions. Next, we formally describe how we model the temporal drift of class prototypes.

**Dynamics Model** We model the evolution of the $K$ prototypes $\mathbf{W}_t = (\mathbf{w}_{t,1}, \ldots, \mathbf{w}_{t,K})$ with $K$ independent Markov processes. The resulting transition model is

$$p(\mathbf{W}_t | \mathbf{W}_{t-1}, \psi^{\text{trans}}) = \prod_{k=1}^K p(\mathbf{w}_{t,k} | \mathbf{w}_{t-1,k}, \psi^{\text{trans}}), \tag{1}$$

where $\psi^{\text{trans}}$ denote the parameters of the transition density. At each time step, the feature vectors $\mathbf{H}_t$ are generated by a mixture distribution over the $K$ classes,

$$p(\mathbf{H}_t | \mathbf{W}_t, \psi^{\text{ems}}) = \prod_{n=1}^{N_t} \sum_{k=1}^K \pi_{t,k} \cdot p(\mathbf{h}_{t,n} | \mathbf{w}_{t,k}, \psi^{\text{ems}}). \tag{2}$$

where $\psi^{\text{ems}}$ are the emission parameters. We thus assume at each time step a standard mixture model over the $K$ classes where the class prototype $\mathbf{w}_{t,k}$ defines the latent class center and $\pi_{t,k}$ the mixture weight for class $k$. The joint distribution of representations, prototypes and class assignments can be factorised as follows,

$$p(\mathbf{H}_{1:T}, \mathbf{W}_{1:T}, \mathbf{C}_{1:T}) = p(\mathbf{W}_1) \prod_{t=2}^T p(\mathbf{W}_t | \mathbf{W}_{t-1}, \psi^{\text{trans}}) \prod_{t=1}^T p(\mathbf{C}_t) p(\mathbf{H}_t | \mathbf{W}_t, \mathbf{C}_t, \psi^{\text{ems}}). \tag{3}$$

We use the notation $\mathbf{H}_{1:T} = \{\mathbf{H}_t\}_{t=1}^T$ to denote the representation vectors $\mathbf{H}_t$ for all time steps $T$ and analogously for $\mathbf{W}_{1:T}$ and $\mathbf{C}_{1:T}$. A plate diagram of the probabilistic model is depicted in App. B. We next outline how we infer the latent class prototypes $\mathbf{W}_{1:T}$.

**Posterior Inference & Adapted Predictions** The primary goal is to update the class prototypes $\mathbf{W}_t$ with the information obtained by the $N_t$ representations of test time $t$. At each test time $t$, we are thus interested in the posterior distribution of the prototypes $p(\mathbf{W}_t | \mathbf{H}_{1:t})$. Once $p(\mathbf{W}_t | \mathbf{H}_{1:t})$ is known, we can update the classification weights with the new posterior mean. We can fit the probabilistic model and infer the posterior distribution for the class weights $\mathbf{W}_t$ and class assignments $\mathbf{C}_t$ with the Expectation-Maximization (EM) algorithm. In the E-step, we compute the posterior $p(\mathbf{W}_{1:T}, \mathbf{C}_{1:T} | \mathbf{H}_{1:T})$. In the M-step, we compute the expectation of the complete-data log-likelihood (Eqn. (3)) with respect to this posterior and then maximize the resulting expression with respect to the model parameters $\phi$:

$$\phi^* = \underset{\phi}{\arg\max} \; \mathbb{E}_{p(\mathbf{W}, \mathbf{C} | \mathbf{H})} \big[ \log p(\mathbf{H}_{1:T}, \mathbf{W}_{1:T}, \mathbf{C}_{1:T}) \big], \tag{4}$$

where $\phi$ comprises the parameters of the transition and emission densities as well as the mixing coefficients, $\phi = \{\psi^{\text{trans}}, \psi^{\text{ems}}, \boldsymbol{\pi}_{1:T}\}$, and we have abbreviated the posterior distribution by $p(\mathbf{W}, \mathbf{C} | \mathbf{H}) :=$

$p(\mathbf{W}_{1:T}, \mathbf{C}_{1:T}|\mathbf{H}_{1:T})$. After one optimization step, we collect the $K$ class prototypes into a matrix $\mathbf{W}_t$. Using the same hidden representations used to fit $\mathbf{W}_t$, we generate the predictions via the model's softmax parameterization,

$$y_{t,n} \sim \text{Cat}\left(y_{t,n}; \text{softmax}(\mathbf{W}_t^\top \mathbf{h}_{t,n})\right), \tag{5}$$

where $y_{t,n}$ denotes a prediction sampled for the representation vector $\mathbf{h}_{t,n}$. Note that adaptation can be performed online by optimizing Eqn. (4) incrementally, considering only data up to point $t$. To omit computing the complete-data log likelihood for an increasing sequence as time goes on, we employ a sliding window approach. Algorithm 1 outlines the overall, procedure of our method. The specific EM updates depend on the chosen parametric form of the SSM. We consider two instances: a Gaussian model and a hyperspherical model based on the von Mises–Fisher distribution. The corresponding EM steps for these two cases are detailed in Algorithm 2 and Algorithm 3, respectively.

---

**Algorithm 1** STAD

1: Input: Source model $f_\theta$, test batches $\mathbf{X}_{1:T}$, sliding window size $s$
2: Initialize: mixing coefficients $\boldsymbol{\pi}_t$, weights $\mathbf{W}_t$, transition and emission parameters $\psi^{\text{trans}}, \psi^{\text{ems}}$
3: **for** $t \in \mathcal{T}$ **do**
4:     Define sliding window: $S_t = \{\tau \mid \max(1, t - s) \leq \tau \leq t\}$
5:     Compute representations: $\mathbf{H}_t = f_\theta^{L-1}(\mathbf{X}_t)$
6:     Fit probabilistic SSM via EM: $\mathbf{W}_t, \mathbf{C}_t, \boldsymbol{\pi}_t, \psi^{\text{trans}}, \psi^{\text{ems}} = \text{EM}\left(\{\mathbf{H}_\tau, \mathbf{W}_\tau, \mathbf{C}_\tau, \boldsymbol{\pi}_\tau\}_{\tau \in S_t}, \psi^{\text{trans}}, \psi^{\text{ems}}\right)$
7:     Predict: $y_{t,n} \sim \text{Cat}\left(y_{t,n}; \text{softmax}(\mathbf{W}_t^\top \mathbf{h}_{t,n})\right)$
8: **end for**

---

**Gaussian Model**   The simplest parametric choice for the transition and emission models is Gaussian. The resulting probabilistic SSM can be interpreted as a mixture of $K$ Kalman filters (KFs) (Kalman, 1960). Owing to the linear–Gaussian structure, the posterior expectation $\mathbb{E}_{p(\mathbf{W},\mathbf{C}|\mathbf{H})}[\cdot]$ in Eqn. (4) can be computed in closed form using the standard KF prediction, update, and smoothing equations (Calabrese & Paninski, 2011; Bishop & Nasrabadi, 2006). The complete model specification is provided in App. B.1, and the corresponding EM updates are summarized in Algorithm 2. However, these closed-form computations come at a cost: they require matrix inversions of size $D \times D$ and the storage of $K \times D^2$ parameters. This makes the Gaussian formulation expensive for high-dimensional features and impractical in low-resource settings. Next, we present a model for spherical features that circumvents these limitations.

### 3.3 Von Mises-Fisher Model for Hyperspherical Features

Choosing Gaussian densities for the transition and emission models, as discussed above, assumes the representation space follows an Euclidean geometry. However, prior work has shown that assuming the hidden representations lie on the unit *hypersphere* results in a better inductive bias for OOD generalization (Mettes et al., 2019; Bai et al., 2024). This is due to the norms of the representations being biased by in-domain information such as class balance, making angular distances a more reliable signal of class membership in the presence of distribution shift (Mettes et al., 2019; Bai et al., 2024). We too employ the hyperspherical assumption by normalizing the hidden representations such that $||\mathbf{h}||_2 = 1$ and model them with the *von Mises-Fisher* (vMF) distribution (Mardia & Jupp, 2009),

$$\text{vMF}(\mathbf{h}; \boldsymbol{\mu}_k, \kappa) = C_D(\kappa) \exp\left\{\kappa \cdot \boldsymbol{\mu}_k^\top \mathbf{h}\right\} \tag{6}$$

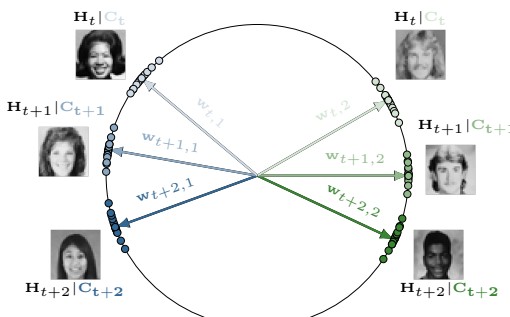

Figure 2: STAD-vMF: Representations lie on the unit sphere. STAD adapts to the distribution shift – induced by changing demographics and styles – by directing the last layer weights $\mathbf{w}_{t,k}$ towards the representations $\mathbf{H}_t$

where $\boldsymbol{\mu}_k \in \mathbb{R}^D$ with $||\boldsymbol{\mu}_k||_2 = 1$ denotes the mean direction of class $k$, $\kappa \in \mathbb{R}^+$ the concentration parameter, and $C_D(\kappa)$ the normalization constant. High values of $\kappa$ imply larger concentration around $\boldsymbol{\mu}_k$. The vMF distribution is proportional to a Gaussian distribution with isotropic variance and unit norm. While previous

work (Mettes et al., 2019; Ming et al., 2023; Bai et al., 2024) has explored training objectives to encourage representations to be vMF-distributed, we apply Eqn. (6) to model the evolving representations.

**Hyperspherical State-Space Model**  Returning to the SSM given above (Eqn. (3)), we specify both transition (Eqn. (1)) and emission models (Eqn. (2)) as vMF distributions, resulting in a hyperspherical transition model, $p(\mathbf{W}_t | \mathbf{W}_{t-1}) = \prod_{k=1}^{K} \mathtt{vMF}(\mathbf{w}_{t,k} | \mathbf{w}_{t-1,k}, \kappa^{\mathrm{trans}})$, and hyperspherical emission model, $p(\mathbf{H}_t | \mathbf{W}_t) = \prod_{n=1}^{N_t} \sum_{k=1}^{K} \pi_{t,k} \mathtt{vMF}(\mathbf{h}_{t,n} | \mathbf{w}_{t,k}, \kappa^{\mathrm{ems}})$. The parameter size of the vMF formulation scales linearly with the feature dimension, $\mathcal{O}(DK)$, compared to the Gaussian case's $\mathcal{O}(D^2 K)$. Notably, the noise parameters, $\kappa^{\mathrm{trans}}, \kappa^{\mathrm{ems}}$ simplify to scalar values which reduces memory substantially. Fig. 2 illustrates this STAD-vMF variant.

**Posterior Inference**  In contrast to the linear Gaussian case, the vMF distribution is not closed under marginalization. As a result, the posterior distribution required for computing the expectation in Eqn. (4) (E-step of the EM algorithm) cannot be expressed in closed form. To address this, we adopt a variational EM approach, approximating the posterior $p(\mathbf{W}, \mathbf{C} \mid \mathbf{H})$ with a mean-field variational distribution $q(\mathbf{W}, \mathbf{C})$ following Gopal & Yang (2014):

$$q(\mathbf{w}_{t,k}) = \mathtt{vMF}(\cdot; \boldsymbol{\rho}_{t,k}, \gamma_{t,k}), \quad q(\mathbf{c}_{t,n}) = \mathtt{Cat}(\cdot; \boldsymbol{\lambda}_{t,n}). \tag{7}$$

The variational distribution $q(\mathbf{W}, \mathbf{C})$ factorizes over $t, n, k$ and the objective from Eqn. (4) becomes $\arg\max_\phi \mathbb{E}_{q(\mathbf{W}, \mathbf{C})}\big[\log p(\mathbf{H}_{1:T}, \mathbf{W}_{1:T}, \mathbf{C}_{1:T})\big]$. The optimal variational parameters are given by

$$\lambda_{t,n,k} = \frac{\pi_{t,k} C_D(\kappa^{\mathrm{ems}}) \exp\left\{\kappa^{\mathrm{ems}} \mathbb{E}_q[\mathbf{w}_{t,k}]^\top \mathbf{h}_{t,n}\right\}}{\sum_{j=1}^{K} \pi_{t,j} C_D(\kappa^{\mathrm{ems}}) \exp\left\{\kappa^{\mathrm{ems}} \mathbb{E}_q[\mathbf{w}_{t,j}]^\top \mathbf{h}_{t,n}\right\}}, \quad \gamma_{t,k} = ||\beta_{t,k}||, \quad \boldsymbol{\rho}_{t,k} = \beta_{t,k}/\gamma_{t,k}, \tag{8}$$

$$\text{with} \quad \beta_{t,k} = \kappa^{\mathrm{trans}} \mathbb{I}_{\{t>1\}} \mathbb{E}_q[\mathbf{w}_{t-1,k}] + \kappa^{\mathrm{ems}} \sum_{n=1}^{N_t} \mathbb{E}_q[c_{t,n,k}] \mathbf{h}_{t,n} + \kappa^{\mathrm{trans}} \mathbb{I}_{\{t<T\}} \mathbb{E}_q[\mathbf{w}_{t+1,k}], \tag{9}$$

where $\mathbb{I}_{\{\cdot\}}$ denotes the indicator function, ensuring that terms are omitted at the temporal boundaries and $\mathbb{E}_q$ denotes expectations with respect to $q(\mathbf{W}, \mathbf{C})$. The expectations of the variational distributions (Eqn. (7)) are $\mathbb{E}_q[\mathbf{c}_{t,n}] = (\lambda_{t,n,1}, \ldots, \lambda_{t,n,K})$ and $\mathbb{E}_q[\mathbf{w}_{t,k}] = A_D(\gamma_{t,k}) \boldsymbol{\rho}_{t,k}$, which completes the E-step. We defer the M-step to App. B.2. Algorithm 3 summarizes the EM algorithm for STAD-vMF. As in the Gaussian case, we form the updated last-layer weight matrix by stacking the posterior means, $\mathbf{W}_t = (\boldsymbol{\rho}_{t,1}, \ldots, \boldsymbol{\rho}_{t,K})$. Notably, posterior inference for the vMF model is much more scalable than the Gaussian case. It operates with linear complexity in $D$, rather than cubic, reducing runtime significantly.

**Recovering the Softmax Predictive Distribution**  In addition to the inductive bias that is beneficial under distribution shift, using the vMF distribution has an additional desirable property: classification via the cluster assignments is equivalent to the original softmax-parameterized classifier. The equivalence is exact under the assumption of equal class proportions and sharing $\kappa$ across classes:

$$\begin{aligned} p(c_{t,n,k} = 1 | \mathbf{h}_{t,n}, \mathbf{w}_{t,1}, \ldots, \mathbf{w}_{t,K}, \kappa^{\mathrm{ems}}) &= \frac{\mathtt{vMF}(\mathbf{h}_{t,n}; \mathbf{w}_{t,k}, \kappa^{\mathrm{ems}})}{\sum_{j=1}^{K} \mathtt{vMF}(\mathbf{h}_{t,n}; \mathbf{w}_{t,j}, \kappa^{\mathrm{ems}})} \\ &= \frac{C_D(\kappa^{\mathrm{ems}}) \exp\left\{\kappa^{\mathrm{ems}} \cdot \mathbf{w}_{t,k}^\top \mathbf{h}_{t,n}\right\}}{\sum_{j=1}^{K} C_D(\kappa^{\mathrm{ems}}) \exp\left\{\kappa^{\mathrm{ems}} \cdot \mathbf{w}_{t,j}^\top \mathbf{h}_{t,n}\right\}} = \mathtt{softmax}\left(\kappa^{\mathrm{ems}} \cdot \mathbf{W}_t^\top \mathbf{h}_{t,n}\right), \end{aligned} \tag{10}$$

which is equivalent to a softmax with temperature-scaled logits, with the temperature set to $1/\kappa^{\mathrm{ems}}$. Temperature scaling only affects the probabilities, not the modal class prediction. If using class-specific $\kappa^{\mathrm{ems}}$ values and assuming imbalanced classes, then these terms show up as class-specific bias terms, $p(c_{t,n,k} = 1 | \mathbf{h}_{t,n}, \mathbf{w}_{t,1}, \ldots, \mathbf{w}_{t,K}, \kappa_1^{\mathrm{ems}}, \ldots, \kappa_K^{\mathrm{ems}}) \propto \exp\left\{\kappa_k^{\mathrm{ems}} \cdot \mathbf{w}_{t,k}^\top \mathbf{h}_{t,n} + \log C_D(\kappa_k^{\mathrm{ems}}) + \log \pi_{t,k}\right\}$.

# 4  Related Work

We overview SSMs and TTA next. App. A provides more detailed discussions on TTA and adjacent fields.

**State-Space Models (SSMs) in Deep Learning** Probabilistic SSMs, such as the Kalman filter (Kalman, 1960), provide a principled framework for updating latent states with new information and have been widely applied in deep learning. In sequence modeling, SSMs are used to learn latent trajectories in both discrete (Krishnan et al., 2015; Karl et al., 2017; Fraccaro et al., 2017; Becker et al., 2019) and continuous time (Schirmer et al., 2022; Ansari et al., 2023; Zhu et al., 2023). Recent advancements in structured SSMs (Gu et al., 2022; Smith et al., 2023; Gu & Dao, 2023) have pushed the state-of-the-art in sequence modeling. However, these models focus on individual sequence dynamics, whereas we are interested in modeling the dynamics of an entire data stream. Our objective aligns more closely with online learning (Duran-Martin et al., 2025). Notably, Chang et al. (2023) and Titsias et al. (2024) extend Kalman filters to handle non-stationary, supervised settings. Like our method, Titsias et al. (2024) infers the evolution of the classification head with a SSM. However, they require labels whereas our method is fully unsupervised.

**Test-Time Adaptation (TTA)** TTA aims to make pre-trained models robust to distribution shifts by adapting directly to the test data during inference. It is training-agnostic, contributing to its growing popularity (Xiao & Snoek, 2024). Early TTA methods recalculate batch normalization (BN) statistics from test data (Nado et al., 2020; Schneider et al., 2020). Updating model parameters via gradient descent is most commonly done via entropy minimization (Grandvalet & Bengio, 2004; Wang et al., 2021; Zhang et al., 2022; Yu et al., 2024; Gao et al., 2024). Alternative approaches include contrastive learning (Chen et al., 2022), invariance regularization (Nguyen et al., 2023), Hebbian learning (Tang et al., 2023), and prompt tuning (Niu et al., 2024). Recent work has focused on making TTA reliably deployable by stress-testing various real-world settings (see App. A for detailed discussions). A key challenge is continual adaptation to a non-stationary target domain, studied in CTTA (Wang et al., 2022b). Solutions include episodic resets (Press et al., 2024), student-teacher models (Döbler et al., 2023; Brahma & Rai, 2023), masking (Liu et al., 2024) and regularization (Niu et al., 2022; Song et al., 2023). Approaches relying on test data statistics further struggle with class imbalance or small test batches, which has led to adaptations in BN strategies (Zhao et al., 2023a; Lim et al., 2023), reservoir sampling (Gong et al., 2022; Yuan et al., 2023), sample filtering (Niu et al., 2023), and label distribution tracking (Zhou et al., 2023). Methods that adapt the classification head, rather than relying on BN, effectively prevent such collapse (Boudiaf et al., 2022; Jang et al., 2023). The most similar approach to ours, T3A (Iwasawa & Matsuo, 2021), recomputes prototypes from representations but relies on heuristics, whereas STAD explicitly models dynamics with a SSM. Lee & Chang (2024b;a); Lee (2025) also use a SSM for online TTA, but to filter noisy model updates, while our SSM models the distribution shift itself. Concurrent to our work, Dai & Yang (2025) proposes Gaussian mixture models updated via EM to adapt the prototypes of vision language models (VLMs). Aside from focusing on VLMs, their model is also static and does not consider transition dynamics.

## 5 Experiments

We evaluate our method, STAD, against various baselines on 7 datasets under challenging settings. Sec. 5.1 studies temporal distribution shifts as defined in Sec. 2, demonstrating the difficulty of the task and STAD's robustness in practical settings. In Sec. 5.2, we go beyond temporal shifts and find that STAD is competitive on reproduction datasets and synthetic corruptions as well (see App. D.1 for results on domain adaptation). Finally, in Sec. 5.3, we provide insights into STAD's mechanisms, confirming the reliability of its prototypes and highlighting the importance of modeling shift dynamics through an ablation study. We now describe the experimental setup. Details are listed in App. C.

**Datasets** We investigate temporal distribution shifts using three image classification datasets spanning several years. Yearbook (Ginosar et al., 2015) involves binary gender prediction on portrait images, capturing changes in demographics and beauty standards over time. EVIS (Zhou et al., 2022a) categorizes vehicles and electronic products. FMoW-Time (Koh et al., 2021) maps satellite images to land use categories. Each dataset comprises samples from multiple years, with the earlier years used for training and the later years for testing (see App. C.1 for details). To evaluate the effectiveness of our method beyond TempTTA, we also test its performance on reproduction datasets (CIFAR-10.1, ImageNetV2) and image corruptions (CIFAR-10-C).

Table 2: Accuracy on **temporal distribution shifts** and label shifts, averaged over three random seeds. Colors highlight performance that either improves or degrades relative to the source model. Best model in bold, second-best underlined.

| Method | Yearbook covariate shift | + label shift | EVIS covariate shift | + label shift | FMoW-Time covariate shift | + label shift |
|---|---|---|---|---|---|---|
| Source model | 81.30 ± 4.18 | | 56.59 ± 0.92 | | 68.94 ± 0.20 | |
| *adapt feature extractor* | | | | | | |
| BN | 84.54 ± 2.10 | 70.47 ± 0.33 | 45.72 ± 2.79 | 14.48 ± 1.02 | 67.60 ± 0.44 | 10.14 ± 0.04 |
| TENT | 84.53 ± 2.11 | 70.47 ± 0.33 | 45.73 ± 2.78 | 14.49 ± 1.02 | 67.86 ± 0.54 | 10.21 ± 0.01 |
| CoTTA | 84.35 ± 2.13 | 66.12 ± 0.87 | 46.13 ± 2.86 | 14.71 ± 1.00 | 68.50 ± 0.25 | 10.19 ± 0.04 |
| SHOT | 85.17 ± 1.89 | 70.71 ± 0.20 | 45.93 ± 2.75 | 14.51 ± 1.00 | 68.02 ± 0.51 | 10.08 ± 0.07 |
| SAR | 84.54 ± 2.10 | 70.47 ± 0.33 | 45.78 ± 2.80 | 14.63 ± 1.00 | 67.87 ± 0.51 | 10.27 ± 0.10 |
| CMF | 85.34 ± 1.86 | 71.20 ± 0.51 | 45.75 ± 3.01 | 35.77 ± 2.07 | 67.44 ± 0.46 | 11.21 ± 0.10 |
| RoTTA | 80.49 ± 3.48 | 80.15 ± 3.50 | 44.28 ± 3.02 | 45.38 ± 2.88 | 67.43 ± 0.67 | 65.77 ± 0.68 |
| *adapt classifier* | | | | | | |
| LAME | 81.60 ± 3.99 | 82.70 ± 4.55 | 56.67 ± 0.99 | **69.37** ± 5.37 | 68.32 ± 0.32 | 83.05 ± 0.48 |
| T3A | 83.49 ± 2.55 | 83.46 ± 2.59 | **57.63** ± 0.77 | 57.32 ± 0.77 | 66.77 ± 0.26 | 66.83 ± 0.27 |
| **STAD-vMF** (ours) | 85.50 ± 1.34 | 84.46 ± 1.19 | 56.67 ± 0.82 | 62.08 ± 1.11 | **68.87** ± 0.06 | **86.25** ± 1.18 |
| **STAD-Gauss** (ours) | **86.22** ± 0.84 | **84.67** ± 1.46 | – | – | – | – |

**Source Architectures and Baselines** We employ a variety of source architectures to demonstrate the model-agnostic nature of STAD. They vary in backbone architecture (ViT, CNN, DenseNet, ResNet, WideResNet) and dimensionality of the representation space (from 32 up to 2048). We list details in App. C.2. We compare against 8 TTA baselines representing fundamental TTA approaches. Six of them adapt the feature extractor: Batch norm adaptation (BN) (Schneider et al., 2020; Nado et al., 2020), TENT (Wang et al., 2021), CoTTA (Wang et al., 2022b), SHOT (Liang et al., 2020) SAR (Niu et al., 2023) and RoTTA (Yuan et al., 2023). Like our method STAD, two baselines adapt the last layer: T3A (Iwasawa & Matsuo, 2021) and LAME (Boudiaf et al., 2022). More details are provided in App. C.3. Batch sizes are the same for all baselines. To ensure optimal performance on newly studied datasets, we conduct an extensive hyperparameter search for each baseline (see App. C.4) and report the best setting.

## 5.1 Temporal Distribution Shifts

We start by evaluating the adaptation abilities to temporal distribution shift on three image classification datasets (Yearbook, EVIS, FMoW-Time), which vary in number of classes (2, 10, 62, respectively), representation dimension (32, 512, 1024, respectively) and shift dynamics (recurring, progressive and rapid, respectively as visible in Fig. 3). For the low-dimensional representations of Yearbook, we also evaluate our computationally costly Gaussian model (STAD-Gauss). We evaluate two settings: **(i) covariate shift with a uniform label distribution** and **(ii) covariate shift with additional shift in the label distribution** $\mathbb{Q}_t(\mathbf{y})$. Having a uniform label distribution—samples are evenly shuffled, making test batches nearly class balanced—has been the traditional evaluation setting for TTA. However, particularly in temporal distribution shifts, it is highly unlikely that samples arrive in this iid-manner. Instead, temporally correlated test streams often observe consecutive samples from the same class (Gong et al., 2022). We follow Lim et al. (2023), ordering the samples by class and thus inducing an extreme label shift. We draw the class order uniformly at random.

**Temporal shifts pose challenges for existing TTA methods.** Tab. 2 shows overall accuracy, averaged over all time steps and three random training seeds. Results that do not outperform the source model are highlighted in red and ones that do in blue. Methods that primarily adapt the feature extractor are shown in the upper section of the table. Ones that, like ours, adapt the classifier are shown in the lower section. To summarize the results: on Yearbook, all methods perform well without label shift, and with label shift, only classifier-based methods improve upon the source baseline. Feature-based methods completely fail on EVIS, and all models, except LAME and STAD-vMF under label shift, fail on FMoW-Time. This leads us to three key takeaways: first, these TempTTA tasks are inherently difficult, leading to smaller adaptation

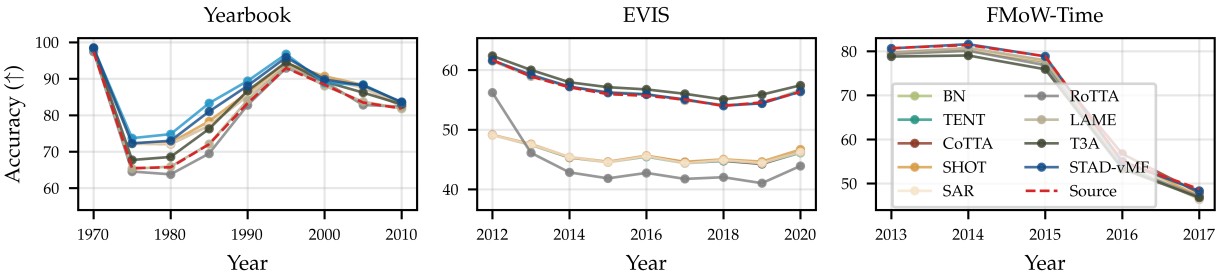

Figure 3: Accuracy over time for TempTTA: STAD mitigates distribution shifts by improving up to 10 points over the source model (Yearbook, 1980s). Some baselines perform similarly, shown by overlaying accuracy trajectories.

gains overall compared to traditional corruption experiments. Second, methods that adapt only the last layer clearly perform better on temporal distribution shifts under both label distribution settings. This indicates that perhaps 'less is more' for TempTTA. Third, STAD demonstrates the most consistent performance, ranking as the best or second-best model across all datasets and settings. On Yearbook, both the Gaussian and vMF variants outperform the baselines, with the fully parameterized Gaussian model better capturing the distribution shift than the more lightweight vMF model. Fig. 3 displays adaptation performance over different timestamps. We see that on EVIS (*middle*) the methods markedly separate, which reflects the aforementioned gap between feature-based and classifier-based approaches. The reader may wonder if STAD can be stacked on a feature-based approach. We present results for BN in Tab. 11 (App. D.3) but find no significant improvement in STAD's performance.

**STAD excels under label shift**  Tab. 2 demonstrates that STAD performs particularly well under imbalanced label distributions, delivering the best results on both Yearbook and FMoW-Time. This advantage stems from STAD's clustering approach, where a higher number of samples from the same ground truth class provides a stronger learning signal, leading to more accurate prototype estimates. This is particularly notable on FMoW, where STAD improves upon the source model by more than 17 points. Further, the performance gap between classifier and feature extractor adaptation methods becomes even more pronounced in this setting. This is not surprising as the latter typically depend heavily on current test-batch statistics, making them vulnerable to imbalanced class distributions (Niu et al., 2023). In contrast, having fewer classes to cause confusion allows classifier-based methods to benefit from label shift, with STAD delivering the most persistent adaptation gains.

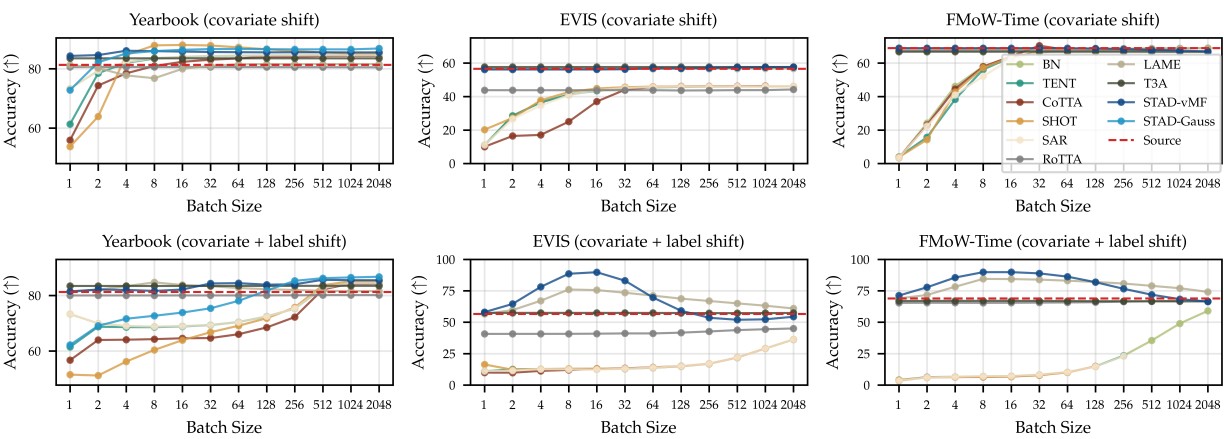

Figure 4: Batch size effects under covariate shift (*first row*) and additional label shift (*second row*): STAD-vMF (dark blue) shows robustness to small batches, with a sweet spot around batch size 16 for label shift on EVIS and FMoW-Time.

**STAD is robust to small batch sizes** Adapting to a small number of samples is crucially valuable, as one does not have to wait for a large batch to accumulate in order to make adapted predictions. We next evaluate performance across 12 different batch sizes ranging from 1 to 2048 under both covariate shift and additional label shift. Fig. 4, displays results. STAD-vMF (dark blue) is able to maintain stable performance under all batch sizes. In Tab. 10 (App. D.2), we report values for batch size 1 showing that STAD adapts successfully even in the most difficult setting. In contrast, methods relying on normalization statistics collapse when not seeing enough samples per adaptation step. For example, on FMoW, feature-based methods collapse to nearly random guessing at the smallest batch sizes. When batch sizes are large, note that TENT, CoTTA, SHOT, and SAR hit memory constraints on FMoW-Time, failing to close the gap to the source model performance in the label shift scenario.

## 5.2 Beyond Temporal Shifts: Reproduction Datasets and Synthetic Corruptions

Although STAD is designed for temporal distribution shifts, we are also interested in the applicability of our method to other types of shifts. Next we report performance on reproduction datasets and synthetic image corruptions.

**Reproduction Datasets** We evaluate our method on reproduction datasets (CIFAR-10.1, ImageNetV2), which have been considered as more realistic and challenging distribution shifts (Zhao et al., 2023b). Tab. 3 confirms the difficulty of adapting to more natural distribution shifts. For CIFAR-10.1, only T3A and STAD outperform the source model for both with and without label shift, with STAD adapting best. For ImageNetV2, none of the methods improve upon the source model when the label distribution is uniform. We again observe that classifier-adaptation methods handle label shifts better by a significant margin.

Table 3: Accuracy on **reproduction datasets**.

| Method | CIFAR-10.1 covariate shift | + label shift | ImageNetV2 covariate shift | + label shift |
|---|---|---|---|---|
| Source model | 88.25 | | 63.18 | |
| *adapt feature extractor* | | | | |
| BN | $86.45 \pm 0.28$ | $23.83 \pm 0.31$ | $62.69 \pm 0.15$ | $43.20 \pm 0.28$ |
| TENT | $86.75 \pm 0.35$ | $23.87 \pm 0.06$ | $63.00 \pm 0.16$ | $43.20 \pm 0.28$ |
| CoTTA | $86.75 \pm 0.17$ | $22.37 \pm 0.25$ | $61.66 \pm 0.29$ | $43.73 \pm 0.33$ |
| SHOT | $86.50 \pm 0.23$ | $23.83 \pm 0.31$ | $62.97 \pm 0.22$ | $43.10 \pm 0.34$ |
| SAR | $86.45 \pm 0.28$ | $23.82 \pm 0.33$ | $62.99 \pm 0.10$ | $43.19 \pm 0.25$ |
| RoTTA | $87.17 \pm 0.21$ | $87.85 \pm 0.35$ | $63.39 \pm 0.20$ | $63.20 \pm 0.21$ |
| *adapt classifier* | | | | |
| LAME | $88.20 \pm 0.09$ | $\mathbf{92.42} \pm 0.28$ | $\mathbf{63.15} \pm 0.10$ | $\underline{80.47} \pm 0.32$ |
| T3A | $\underline{88.28} \pm 0.06$ | $89.00 \pm 0.66$ | $62.86 \pm 0.04$ | $63.47 \pm 0.09$ |
| **STAD-vMF** (ours) | $\mathbf{88.42} \pm 0.10$ | $\underline{92.23} \pm 0.70$ | $62.39 \pm 0.05$ | $\mathbf{81.46} \pm 0.24$ |

**Synthetic Corruptions** Lastly, we test our method on gradually increasing noise corruptions of CIFAR-10-C, a standard TTA benchmark and provide results on ImageNet-C in App. D.4. Tab. 4 shows the accuracy averaged across all corruption types for CIFAR-10-C. We make three key observations. First, performance gains are much higher than on previous datasets indicating the challenge posed by non-synthetic shifts. Second, as expected, methods adapting the backbone model are more performative on input-level noise, since such shifts primarily affect earlier layers (Tang et al., 2023; Lee et al., 2023). Lastly, STAD is consistently the best method amongst those adapting only the last layer.

Table 4: Accuracy on **synthetic corruptions** (CIFAR-10-C)

| Method | Corruption severity 1 | 2 | 3 | 4 | 5 | Mean |
|---|---|---|---|---|---|---|
| Source | 86.90 | 81.34 | 74.92 | 67.64 | 56.48 | 73.46 |
| *adapt feature extractor* | | | | | | |
| BN | 90.18 | 88.16 | 86.24 | 83.18 | 79.27 | 85.41 |
| TENT | **90.87** | **89.70** | 88.32 | 85.89 | 83.09 | 87.57 |
| CoTTA | 90.62 | 89.42 | **88.55** | **87.28** | **85.27** | **88.23** |
| SHOT | 90.31 | 88.66 | 87.31 | 85.02 | 82.13 | 86.69 |
| SAR | 90.16 | 88.09 | 86.26 | 83.32 | 79.48 | 85.46 |
| RoTTA | 90.60 | 89.41 | 88.14 | 85.88 | 83.37 | 87.48 |
| *adapt classifier* | | | | | | |
| LAME | 86.94 | 81.39 | 74.93 | 67.69 | 56.47 | 73.48 |
| T3A | 87.83 | 82.75 | 76.77 | 69.43 | 57.90 | 74.94 |
| **STAD-vMF** (ours) | 88.21 | 83.68 | 78.42 | 72.19 | 62.44 | 76.99 |

## 5.3 Analysis of Tracking Abilities

Lastly, we seek to further understand the reasons behind STAD's strong performance. At its core, STAD operates through a mechanism of *dynamic clustering*. We next inspect the importance of STAD's dynamics component and assess the fidelity of its clustering.

**Clusters are reliable.** We evaluate how well STAD's inferred cluster centers align with the ground truth cluster centers (computed using labels). We chose the progressively increasing distribution shift of CIFAR-10-C as this dataset represents a bigger challenge for STAD. Fig. 5 *(left)* shows distance (in angular degrees) to the ground truth cluster centers for both the source model and STAD. STAD (blue line) adapts effectively, significantly reducing the angular distance to the ground truth cluster centers. For the source model, the progressive distribution shift causes the ground truth cluster centers to drift increasingly further from the source prototypes (red line). Additionally, by computing dispersion (Ming et al., 2023) (Fig. 5, *middle*), which measures the spread of the prototypes (in angular degrees), we find that STAD mirrors the ground truth trend (yellow line) of clusters becoming closer together. This is a promising insight, as it suggests that STAD's cluster dispersion could potentially serve as an unsupervised metric to proactively flag when clusters start overlapping and estimate adaptation accuracy. In Fig. 5 *(right)*, we plot accuracy vs dispersion of STAD's prototypes for different corruptions and severity levels, confirming that they positively correlate.

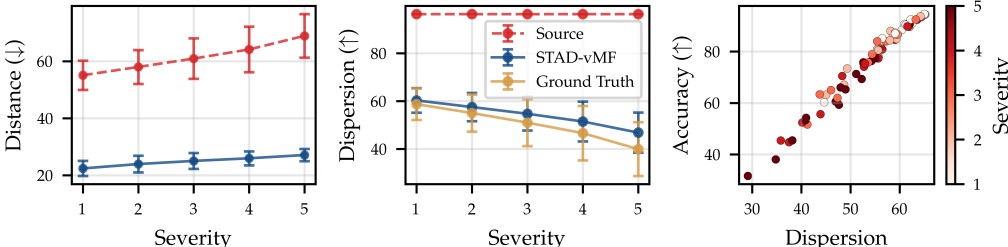

Figure 5: Cluster fidelity on CIFAR-10-C

**Dynamics are crucial.** STAD is proposed with the assumption that adapting the class prototypes based on those of the previous time step facilitates rapid and reliable adaptation. However, one could also consider a static version of STAD that does not have a transition model (Eqn. (1)). Rather, the class prototypes are computed as a standard mixture model (Eqn. (2)) without considering previously inferred prototypes. Tab. 5 presents the accuracy differences between the static and dynamic versions of STAD. Removing STAD's transition model results in a *substantial performance drop* of up to 28 percentage points. This supports our assumption that SSMs are well-suited for TempTTA.

Table 5: Accuracy of dynamic and static versions of STAD (i.e. when removing the transition model)

| Variant | Yearbook | FMoW | CIFAR-10-C |
|---|---|---|---|
| STAD-vMF with dynamics | **85.50** $\pm$ 1.30 | **86.25** $\pm$ 1.18 | **76.99** |
| STAD-vMF w/o dynamics | 61.03 $\pm$ 2.92 | 68.87 $\pm$ 0.28 | 73.57 |
| Delta | −24.47 | −17.38 | −3.41 |
| STAD-Gauss with dynamics | **86.22** $\pm$ 0.84 | – | – |
| STAD-Gauss w/o dynamics | 57.79 $\pm$ 2.14 | – | – |
| Delta | −28.43 | – | – |

## 6 Discussion and Conclusion

We studied temporal distribution shifts and demonstrated the significant challenges they pose for existing TTA methods. We proposed STAD, a novel TTA strategy based on probabilistic state-space models to address temporal shifts. Our Gaussian and vMF variants of STAD effectively track the evolution of the last linear layer under distribution shifts, enabling unsupervised adaptation in deployed models. We found STAD to be most effective for structural temporal shifts and label shifts (Tabs. 2 and 3). While effective across a range of settings, STAD's design inherits limitations regarding the types of distribution shift it can mitigate. Notably, its performance depends on the shift being visible in the last layer, as adapting

only the final classifier is less effective when earlier layers are primarily affected—a behavior we observe for synthetic corruptions in CIFAR-10-C (Table 4) and ImageNet-C (Table 12), consistent with prior work (Lee et al., 2023). STAD also assumes that last-layer representations change gradually over time, and that class prototypes remain temporally correlated. This makes abrupt shifts more challenging to address, though in our experiments we did not encounter a scenario where sudden shifts fully broke the temporal correlation (see App. D.1). Future work on TempTTA could incorporate timestamps to better model time progression or determine when adaptation is no longer feasible.

## Broader Impact Statement

This work introduces a probabilistic state-space framework for TTA under temporal distribution shifts. By modeling time-evolving feature dynamics and inferring class prototypes without labels, STAD enables continuous model adaptation in settings where labels are unavailable. STAD contributes to more robust and flexible deployment of machine learning systems in non-stationary environments. However, because the method operates without ground-truth labels, there is a risk of compounding errors if the inferred prototypes drift too far from reality—especially in cases of abrupt or adversarial shifts. Overreliance on unsupervised adaptation without safeguards could lead to silent model degradation. Practitioners should therefore monitor model behavior carefully and understand the assumptions underlying the temporal dynamics. Overall, this work supports the development of adaptive models under distribution shift.

## Acknowledgment

We thank Metod Jazbec for valuable discussions and helpful feedback on the draft, and we thank Rajeev Verma for constructive comments on an early manuscript. This project was generously supported by the Bosch Center for Artificial Intelligence. Eric Nalisnick did not utilize resources from Johns Hopkins University for this project.

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

# A    Appendix

## Appendix

The appendix is structured as follows:

- App. A discusses further related work such as adjacent robustness paradigms and further TTA settings that aim for realistic evaluations.

- App. B provides methodological details on STAD.

  - App. B.1 defines the Gaussian SSM (STAD-Gauss), provides the respective EM update equations and an algorithmic overview (Algorithm 2).
  - App. B.2 details the von Mises-Fisher SSM (STAD-vMF). Here, we notably provide the inference scheme including all update equations for the variational EM step as well as an algorithmic summary (Algorithm 3).

- App. C contains various implementation details.

- App. D provides additional experimental results on

  - domain adaptation benchmarks (App. D.1)
  - single sample adaptation (App. D.2)
  - STAD in combination with BN (App. D.3)
  - ImageNet-C (App. D.4)
  - comparison to a supervised oracle (App. D.5)
  - visualizations of representation space (App. D.6)
  - runtime comparisons (App. D.7)
  - sensitivity to hyperparameters (App. D.8)

# A   Expanded Related Work

Table 6: Comparison of realistic TTA settings and associated methods. We distinguish settings with respect to assumptions on the feature distribution and label distribution as well as if inherently time-ordered data streams and single sample settings have been considered. Notably, we categorize the assumption on the feature distribution into *stationary* (a single, static feature distribution), *non-stationary* (continually evolving distribution) and *mixture* (mixture distribution, e.g. samples of different corruption types observed simultaneously).

| TTA setting | Related method | Feature distribution $\mathbb{Q}(\mathbf{x})$ | Label distribution $\mathbb{Q}(y)$ | Time-ordered | Single sample |
|---|---|---|---|---|---|
| Fully TTA | TENT (Wang et al., 2021) | stationary | balanced | ✗ | ✗ |
| Continual TTA | CoTTA (Wang et al., 2022b) | non-stationary | balanced | ✗ | ✗ |
| Non-iid TTA | LAME (Boudiaf et al., 2022) | stationary | imbalanced | ✗ | ✗ |
| | NOTE (Gong et al., 2022) | stationary | imbalanced | ✓ | ✗ |
| Practical TTA | RoTTA (Yuan et al., 2023) | non-stationary | imbalanced | ✗ | ✗ |
| Wild TTA | SAR (Niu et al., 2023) | mixture | imbalanced | ✗ | ✓ |
| N/A | TTN (Lim et al., 2023) | stationary, non-stationary, mixture | imbalanced | ✗ | ✓ |
| N/A | RMT (Döbler et al., 2023) | non-stationary | balanced | ✗ | ✓ |
| Universal TTA | ROID (Marsden et al., 2024) | non-stationary, mixture | imbalanced | ✗ | ✓ |
| Real-world TTA | TRIBE (Su et al., 2024) | non-stationary | imbalanced | ✗ | ✗ |
| UniTTA | UniTTA (Du et al., 2025) | non-stationary, mixture | imbalanced | ✗ | ✗ |
| TempTTA | STAD (ours) | non-stationary | imbalanced | ✓ | ✓ |

**Realistic TTA** aims to evaluate models under challenging test conditions that may occur in real-world deployments. We provide a tabular overview of prominent and recent work in Tab. 6. For a more comprehensive review of TTA settings and methods, see Xiao & Snoek (2024). Since TTA methods typically rely on test data statistics, they are often sensitive to the ordering of samples in the test data stream. As a result, prior work has explored the robustness of TTA under different sample ordering strategies. Two key factors influence sample order: the domain index, which determines the feature distribution $\mathbb{Q}(\mathbf{x})$, and the class index, which determines the label distribution $\mathbb{Q}(y)$.

In the standard *fully TTA* setting (Wang et al., 2021), the test stream consists of a single domain, and test data is sampled i.i.d., resulting in a uniform label distribution per batch. *Continual TTA* (Wang et al., 2022b) extends this setting by considering multiple domains sequentially, leading to a non-stationary feature distribution. *Non-i.i.d. TTA* (Boudiaf et al., 2022; Gong et al., 2022; Zanella et al., 2025) is another extension of fully TTA that challenges the i.i.d. assumption by introducing temporal correlation in the sampling procedure, causing an imbalanced label distribution per test batch. The *Practical TTA* setting (Yuan et al., 2023) combines continual TTA and non-i.i.d. TTA by incorporating both continually changing domains and temporal correlation. *Wild TTA* (Niu et al., 2023) is another extension of non-i.i.d. TTA that additionally studies mixed domains (see also *Universal TTA* (Marsden et al., 2024)) and single sample adaptation (see also Lim et al. (2023); Döbler et al. (2023); Marsden et al. (2024)). *Real-world TTA* (Su et al., 2024) builds on practical TTA by also controlling the global label distribution across the data stream. Lastly, *UniTTA* (Du et al., 2025) comprises 36 sampling strategies, considering ordering and imbalance of both domains and class labels.

While past work has studied adaptation to non-stationary distributions, they mostly consider a sequence of categorical domains (e.g. different corruption types). In contrast, TempTTA focuses on sequences where the domain index is temporal. The data stream thus follows an inherent time-ordering (Bai et al., 2023). To test the applicablity of our method in challenging settings, we also evaluate single sample adaptation and class-imbalanced label distributions in Sec. 5.1.

**Domain Generalization (DG)** The goal of DG (Zhou et al., 2022a) is to learn a predictive model that can generalize well to any unseen domains, assuming access to multiple source domains at the training stage. Examples of common approaches are domain-invariant feature learning (Arjovsky et al., 2019), data augmentation (Zhang, 2018) and regularization (Krueger et al., 2021). The most relevant subfield to this work is temporal DG (TDG) (Nasery et al., 2021; Qin et al., 2022; Bai et al., 2023; Xie et al., 2024b; Jin et al., 2024; Zeng et al., 2024; Cai et al., 2024; Xie et al., 2024a), which models dynamics from sequential source

domains to generalize to evolving target domains. While both TDG and TempTTA address temporal shifts, TDG operates during training and requires a sequence of labeled source domains, whereas our approach improves the performance of arbitrary pre-trained models at test time using only a stream of unlabeled data.

**Unsupervised Domain Adaptation (UDA)**    UDA improves generalization by exploiting both labeled source data and unlabeled target data. Most related to our setting is gradual domain adaptation (GDA) (Hoffman et al., 2014; Wulfmeier et al., 2018; Bobu et al., 2018; Kumar et al., 2020; Abnar et al., 2021; Wang et al., 2022a), which aims to adapt to a target domain by exploiting intermediate domains with gradual distribution shift between source and target. GDA methods rely on access to both source and target data for distribution alignment. However, in many practical scenarios, source data may be unavailable due to e.g. privacy concerns, motivating the need for TTA.

## B    Methodological Details

In the following, we provide details on the probabilistic model and inference of STAD. Fig. 6 displays the plate diagram.

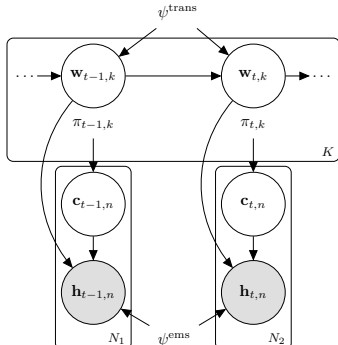

Figure 6: Graphical Model: Representations $\mathbf{h}_{t,n}$ are modeled with a dynamic mixture model. Latent class prototypes $\mathbf{w}_{t,k}$ evolve at each time step, cluster assignments $\mathbf{c}_{t,n}$ determine class membership.

### B.1    Details on STAD-Gaussian

**Gaussian State-Space Model**    We use a linear Gaussian transition model to describe the weight evolution over time: For each class $k$, the weight vector evolves according to a linear drift parameterized by a class-specific transition matrix $\mathbf{A}_k \in \mathbb{R}^{D \times D}$. This allows each class to have independent dynamics. The transition noise follows a multivariate Gaussian distribution with zero mean and global covariance $\boldsymbol{\Sigma}^{\text{trans}} \in \mathbb{R}^{D \times D}$. The transition noise covariance matrix is a shared parameter across classes and time points to prevent overfitting and keep the parameter size manageable. Eqn. (11) states the Gaussian transition density.

$$\text{Transition model:} \quad p(\mathbf{W}_t|\mathbf{W}_{t-1}) = \prod_{k=1}^{K} \mathcal{N}(\mathbf{w}_{t,k}|\mathbf{A}_k\mathbf{w}_{t-1,k}, \boldsymbol{\Sigma}^{\text{trans}}) \tag{11}$$

$$\text{Emission model:} \quad p(\mathbf{H}_t|\mathbf{W}_t) = \prod_{n=1}^{N_t} \sum_{k=1}^{K} \pi_{t,k} \mathcal{N}(\mathbf{h}_{t,n}|\mathbf{w}_{t,k}, \boldsymbol{\Sigma}^{\text{ems}}) \tag{12}$$

Eqn. (12) gives the emission model of the observed features $\mathbf{H}_t$ at time $t$. As in Eqn. (2), the features at a given time $t$ are generated by a mixture distribution with mixing coefficient $\pi_{t,k}$. The emission density of each of the $K$ components is a multivariate normal with the weight vector of class $k$ at time $t$ as mean and $\boldsymbol{\Sigma}^{\text{ems}} \in \mathbb{R}^{D \times D}$ as class-independent covariance matrix. The resulting model can be seen as a mixture of $K$ Kalman filters. Variants of it have found application in applied statistics (Calabrese & Paninski, 2011).

**Posterior inference** We use the EM objective of Eqn. (4) to maximize for the model parameters $\phi = \{\{\mathbf{A}_k, \{\pi_{t,k}\}_{t=1}^T\}_{k=1}^K, \mathbf{\Sigma}^{\text{trans}}, \mathbf{\Sigma}^{\text{ems}}\}$. Thanks to the linearity and Gaussian assumptions, the posterior expectation $\mathbb{E}_{p(\mathbf{W},\mathbf{C}|\mathbf{H})}[\cdot]$ in Eqn. (4) (E-step) can be computed analytically using the well-known Kalman filter predict, update, and smoothing equations (Calabrese & Paninski, 2011; Bishop & Nasrabadi, 2006) as outlined next.

**E-step** The posterior responsibilities of each component are given by the Gaussian mixture posterior:

$$\mathbb{E}[c_{t,n,k}] = \frac{\pi_{t,k}|\mathbf{\Sigma}^{\text{ems}}|^{-1/2}\exp\left\{-\frac{1}{2}(\mathbf{h}_{t,n} - \mathbf{w}_{t,k})^\top(\mathbf{\Sigma}^{\text{ems}})^{-1}(\mathbf{h}_{t,n} - \mathbf{w}_{t,k})\right\}}{\sum_{j=1}^K \pi_{t,j}|\mathbf{\Sigma}^{\text{ems}}|^{-1/2}\exp\left\{-\frac{1}{2}(\mathbf{h}_{t,n} - \mathbf{w}_{t,j})^\top(\mathbf{\Sigma}^{\text{ems}})^{-1}(\mathbf{h}_{t,n} - \mathbf{w}_{t,j})\right\}}, \tag{13}$$

which we summarize as $\mathbb{E}[c_{t,n,k}] = \text{computeAssignments}(\mathbf{h}_{t,n}, \mathbf{w}_{t,k}, \pi_{t,k}, \mathbf{\Sigma}^{\text{ems}})$. Given the previous posterior $(\boldsymbol{\mu}_{t-1,k}^+, \mathbf{\Sigma}_{t-1,k}^+)$, the prior distribution at time $t$ follows the standard Kalman filter prediction:

$$\boldsymbol{\mu}_{t,k}^- = \mathbf{A}_k\boldsymbol{\mu}_{t-1,k}^+, \quad \mathbf{\Sigma}_{t,k}^- = \mathbf{\Sigma}^{\text{trans}} + \mathbf{A}_k\mathbf{\Sigma}_{t-1,k}^+\mathbf{A}_k^\top. \tag{14}$$

We denote this step as $(\boldsymbol{\mu}_{t,k}^-, \mathbf{\Sigma}_{t,k}^-) = \text{predict}(\mathbf{A}_k, \mathbf{\Sigma}^{\text{trans}}, \boldsymbol{\mu}_{t-1,k}^+, \mathbf{\Sigma}_{t-1,k}^+)$. Conditioning the predicted state on the current observation $\mathbf{h}_{t,n}$ yields:

$$\boldsymbol{\mu}_{t,n,k}^+ = \mathbf{\Sigma}_{t,n,k}^+\left[(\mathbf{\Sigma}_{t,k}^-)^{-1}\boldsymbol{\mu}_{t,k}^- + \mathbb{E}[c_{t,n,k}](\mathbf{\Sigma}^{\text{ems}})^{-1}\mathbf{h}_{t,n}\right], \tag{15}$$

$$\mathbf{\Sigma}_{t,n,k}^+ = \left[(\mathbf{\Sigma}_{t,k}^-)^{-1} + \mathbb{E}[c_{t,n,k}](\mathbf{\Sigma}^{\text{ems}})^{-1}\right]^{-1}, \tag{16}$$

that is, $(\boldsymbol{\mu}_{t,n,k}^+, \mathbf{\Sigma}_{t,n,k}^+) = \text{update}(\boldsymbol{\mu}_{t,k}^-, \mathbf{\Sigma}_{t,k}^-, \mathbf{h}_{t,n}, \mathbb{E}[c_{t,n,k}])$. This corresponds to a Kalman filter trajectory for each observation and class. In order to aggregate the observation-wise posterior to one single posterior over each class, we chose a mixture distribution with weights $\alpha_{t,n,k} = \frac{\mathbb{E}[c_{t,n,k}]}{\sum_{i=1}^{N_t}\mathbb{E}[c_{t,i,k}]}$ which gives:

$$\boldsymbol{\mu}_{t,k}^+ = \sum_{n=1}^{N_t}\alpha_{t,n,k}\boldsymbol{\mu}_{t,n,k}^+, \quad \mathbf{\Sigma}_{t,k}^+ = \sum_{n=1}^{N_t}\alpha_{t,n,k}\mathbf{\Sigma}_{t,n,k}^+ + \sum_{n=1}^{N_t}\alpha_{t,n,k}(\boldsymbol{\mu}_{t,n,k}^+ - \boldsymbol{\mu}_{t,k}^+)(\boldsymbol{\mu}_{t,n,k}^+ - \boldsymbol{\mu}_{t,k}^+)^\top, \tag{17}$$

or compactly $(\boldsymbol{\mu}_{t,k}^+, \mathbf{\Sigma}_{t,k}^+) = \text{mixture}(\{\boldsymbol{\mu}_{t,n,k}^+, \mathbf{\Sigma}_{t,n,k}^+, \alpha_{t,n,k}\}_{n=1}^{N_t})$. Smoothing backward in time gives:

$$\boldsymbol{\mu}_{t,k} = \boldsymbol{\mu}_{t,k}^+ + \mathbf{J}_{t,k}\left(\boldsymbol{\mu}_{t+1,k} - \mathbf{A}_k\boldsymbol{\mu}_{t,k}^+\right), \quad \mathbf{\Sigma}_{t,k} = \mathbf{\Sigma}_{t,k}^+ + \mathbf{J}_{t,k}\left(\mathbf{\Sigma}_{t+1,k} - \mathbf{\Sigma}_{t,k}^-\right)\mathbf{J}_{t,k}^\top, \tag{18}$$

$$\mathbf{J}_{t,k} = \mathbf{\Sigma}_{t,k}^+\mathbf{A}_k^\top(\mathbf{\Sigma}_{t,k}^-)^{-1}, \tag{19}$$

i.e., $(\boldsymbol{\mu}_{t,k}, \mathbf{\Sigma}_{t,k}) = \text{smooth}(\boldsymbol{\mu}_{t,k}^+, \mathbf{\Sigma}_{t,k}^+, \boldsymbol{\mu}_{t+1,k}, \mathbf{\Sigma}_{t+1,k}, \mathbf{\Sigma}_{t,k}^-, \mathbf{A}_k)$. Finally, from the smoothed posteriors we compute the necessary expectations required in the M-step:

$$\mathbb{E}[\mathbf{w}_{t,k}] = \boldsymbol{\mu}_{t,k}, \quad \mathbb{E}[\mathbf{w}_{t,k}\mathbf{w}_{t-1,k}^\top] = \mathbf{\Sigma}_{t,k}\mathbf{J}_{t-1,k}^\top + \boldsymbol{\mu}_{t,k}\boldsymbol{\mu}_{t-1,k}^\top, \quad \mathbb{E}[\mathbf{w}_{t,k}\mathbf{w}_{t,k}^\top] = \mathbf{\Sigma}_{t,k} + \boldsymbol{\mu}_{t,k}\boldsymbol{\mu}_{t,k}^\top, \tag{20}$$

in short, $\{\mathbb{E}[\mathbf{w}_{t,k}], \mathbb{E}[\mathbf{w}_{t,k}\mathbf{w}_{t-1,k}^\top], \mathbb{E}[\mathbf{w}_{t,k}\mathbf{w}_{t,k}^\top]\} = \text{computeMarginals}(\boldsymbol{\mu}_{t,k}, \mathbf{\Sigma}_{t,k}, \boldsymbol{\mu}_{t-1,k}, \mathbf{J}_{t-1,k})$.

**M-step** Given the responsibilities $\mathbb{E}[c_{t,n,k}]$ and the smoothed posteriors of the prototypes $(\boldsymbol{\mu}_{t,k}, \mathbf{\Sigma}_{t,k})$, the parameters are re-estimated by maximizing the expected complete-data log-likelihood.

The posterior mixture weights are given by the average responsibilities:

$$\pi_{t,k} = \frac{1}{N_t}\sum_{n=1}^{N_t}\mathbb{E}[c_{t,n,k}], \tag{21}$$

i.e., $\pi_{t,k} = \text{updatePi}(\{\mathbb{E}[c_{t,n,k}]\}_{n=1}^{N_t})$. Maximizing the likelihood of the linear-Gaussian state evolution yields the least-squares estimate of the transition matrices:

$$\mathbf{A}_k = \left(\sum_{t=2}^T\mathbb{E}[\mathbf{w}_{t,k}\mathbf{w}_{t-1,k}^\top]\right)\left(\sum_{t=2}^T\mathbb{E}[\mathbf{w}_{t-1,k}\mathbf{w}_{t-1,k}^\top]\right)^{-1}, \tag{22}$$

or equivalently $\mathbf{A}_k = \text{updateA}(\{\mathbb{E}[\mathbf{w}_{t,k}\mathbf{w}_{t-1,k}^\top], \mathbb{E}[\mathbf{w}_{t-1,k}\mathbf{w}_{t-1,k}^\top]\}_{t=2}^T)$. The transition noise covariance is obtained by the maximum-likelihood estimator of the process noise:

$$\boldsymbol{\Sigma}^{\text{trans}} = \frac{1}{(T-1)K} \sum_{k=1}^K \sum_{t=2}^T \Big( \mathbb{E}[\mathbf{w}_{t,k}\mathbf{w}_{t,k}^\top] - \mathbb{E}[\mathbf{w}_{t,k}\mathbf{w}_{t-1,k}^\top]\mathbf{A}_k^\top - \mathbf{A}_k\mathbb{E}[\mathbf{w}_{t-1,k}\mathbf{w}_{t,k}^\top] + \mathbf{A}_k\mathbb{E}[\mathbf{w}_{t-1,k}\mathbf{w}_{t-1,k}^\top]\mathbf{A}_k^\top \Big),$$

(23)

in short, $\boldsymbol{\Sigma}^{\text{trans}} = \text{updateSigmaTrans}(\{\mathbb{E}[\mathbf{w}_{t,k}\mathbf{w}_{t,k}^\top], \mathbb{E}[\mathbf{w}_{t,k}\mathbf{w}_{t-1,k}^\top], \mathbb{E}[\mathbf{w}_{t-1,k}\mathbf{w}_{t-1,k}^\top], \mathbf{A}_k\}_{t=2}^T)$. Finally, maximizing the likelihood of the Gaussian emission model yields the responsibility-weighted residual covariance:

$$\boldsymbol{\Sigma}^{\text{ems}} = \frac{\sum_{k=1}^K \sum_{t=1}^T \sum_{n=1}^{N_t} \mathbb{E}[c_{t,n,k}]\big(\mathbf{h}_{t,n}\mathbf{h}_{t,n}^\top - \mathbb{E}[\mathbf{w}_{t,k}]\mathbf{h}_{t,n}^\top - \mathbf{h}_{t,n}\mathbb{E}[\mathbf{w}_{t,k}]^\top + \mathbb{E}[\mathbf{w}_{t,k}\mathbf{w}_{t,k}^\top]\big)}{\sum_{k=1}^K \sum_{t=1}^T \sum_{n=1}^{N_t} \mathbb{E}[c_{t,n,k}]},$$

(24)

or compactly $\boldsymbol{\Sigma}^{\text{ems}} = \text{updateSigmaEms}(\{\mathbf{h}_{t,n}, \mathbb{E}[c_{t,n,k}], \mathbb{E}[\mathbf{w}_{t,k}], \mathbb{E}[\mathbf{w}_{t,k}\mathbf{w}_{t,k}^\top]\}_{t,n,k})$.

**Complexity** The closed-form computations of the posterior $p(\mathbf{W}_t|\mathbf{H}_{1:t})$ and smoothing $p(\mathbf{W}_t|\mathbf{H}_{1:T})$ densities come at a cost as they involve, amongst others, matrix inversions of dimensionality $D \times D$. This results in considerable computational costs and can lead to numerical instabilities when feature dimension $D$ is large. In addition, the parameter size scales as $K \times D^2$, risking overfitting and consuming substantial memory. These are limitations of the Gaussian formulation, making it costly for high-dimensional feature spaces and impractical in low-resource environments requiring instant predictions.

---

**Algorithm 2** EM for STAD-Gauss

---

1: Input: $\{\mathbf{W}_\tau, \mathbf{C}_\tau, \boldsymbol{\pi}_\tau, \mathbf{H}_\tau\}_{\tau \in S_t}$, $\boldsymbol{\Sigma}^{\text{trans}}, \mathbf{A}_k, \boldsymbol{\Sigma}^{\text{ems}}$
2: **for** $\tau \in S_t$ **do**
3:     *E-step*
4:     $\mathbb{E}[c_{\tau,n,k}] = \text{computeAssignments}(\mathbf{h}_{\tau,n}, \mathbf{w}_{\tau,k}, \pi_{\tau,k}, \boldsymbol{\Sigma}^{\text{ems}})$
5:     $\boldsymbol{\mu}_{\tau,k}^-, \boldsymbol{\Sigma}_{\tau,k}^- = \text{predict}(\mathbf{A}_k, \boldsymbol{\Sigma}^{\text{trans}}, \boldsymbol{\mu}_{\tau-1,k}^+, \boldsymbol{\Sigma}_{\tau-1,k}^+)$
6:     $\boldsymbol{\mu}_{\tau,n,k}^+, \boldsymbol{\Sigma}_{\tau,n,k}^+ = \text{update}(\boldsymbol{\mu}_{\tau,k}^-, \boldsymbol{\Sigma}_{\tau,k}^-, \mathbf{h}_{\tau,n}, \mathbb{E}[c_{\tau,n,k}])$
7:     $\boldsymbol{\mu}_{\tau,k}^+, \boldsymbol{\Sigma}_{\tau,k}^+ = \text{mixture}(\{\boldsymbol{\mu}_{\tau,n,k}^+, \boldsymbol{\Sigma}_{\tau,n,k}^+, \mathbb{E}[c_{\tau,n,k}]\}_{n=1}^{N_\tau})$
8:     $\boldsymbol{\mu}_{\tau,k}, \boldsymbol{\Sigma}_{\tau,k} = \text{smooth}(\boldsymbol{\mu}_{\tau,k}^+, \boldsymbol{\Sigma}_{\tau,k}^+, \boldsymbol{\mu}_{\tau+1,k}, \boldsymbol{\Sigma}_{\tau+1,k}, \mathbf{A}_k)$
9:     $\mathbb{E}[\mathbf{w}_{\tau,k}], \mathbb{E}[\mathbf{w}_{\tau,k}\mathbf{w}_{\tau-1,k}^\top], \mathbb{E}[\mathbf{w}_{\tau,k}\mathbf{w}_{\tau,k}^\top] = \text{computeMarginals}(\boldsymbol{\mu}_{\tau,k}, \boldsymbol{\Sigma}_{\tau,k}, \boldsymbol{\mu}_{\tau-1,k}, \mathbf{J}_{\tau-1,k})$
10:     *M-step*
11:     $\pi_{\tau,k} = \text{updatePi}(\{\mathbb{E}[c_{\tau,n,k}]\}_{n=1}^{N_\tau})$
12: **end for**
13: $\mathbf{A}_k = \text{updateA}(\{\mathbb{E}[\mathbf{w}_{\tau,k}\mathbf{w}_{\tau-1,k}^\top], \mathbb{E}[\mathbf{w}_{\tau-1,k}\mathbf{w}_{\tau-1,k}^\top]\}_{\tau \in S_t, \ \tau > \min S_t})$
14: $\boldsymbol{\Sigma}^{\text{trans}} = \text{updateSigmaTrans}(\{\mathbb{E}[\mathbf{w}_{\tau,k}\mathbf{w}_{\tau,k}^\top], \mathbb{E}[\mathbf{w}_{\tau,k}\mathbf{w}_{\tau-1,k}^\top], \mathbb{E}[\mathbf{w}_{\tau-1,k}\mathbf{w}_{\tau-1,k}^\top], \mathbf{A}_k\}_{\tau \in S_t, \ \tau > \min S_t})$
15: $\boldsymbol{\Sigma}^{\text{ems}} = \text{updateSigmaEms}(\{\mathbf{h}_{\tau,n}, \mathbb{E}[c_{\tau,n,k}], \mathbb{E}[\mathbf{w}_{\tau,k}], \mathbb{E}[\mathbf{w}_{\tau,k}\mathbf{w}_{\tau,k}^\top]\}_{\tau,n,k})$
16: **return** $\mathbf{W}_t, \mathbf{C}_t, \boldsymbol{\pi}_t, \boldsymbol{\Sigma}^{\text{trans}}, \boldsymbol{\Sigma}^{\text{ems}}$

---

## B.2 Details on STAD-vMF

**Complete-data log likelihood**   Using the von Mises–Fisher distribution as the hyperspherical transition and emission model, the log of the complete-data likelihood in Eqn. (3) becomes

$$\log p(\mathbf{H}_{1:T}, \mathbf{W}_{1:T}, \mathbf{C}_{1:T}) = \sum_{k=1}^{K} \log p(\mathbf{w}_{1,k}) \tag{25}$$

$$+ \sum_{t=1}^{T} \sum_{n=1}^{N_t} \log p(\mathbf{c}_{t,n}) + \sum_{k=1}^{K} c_{t,n,k} \log p(\mathbf{h}_{t,n} \mid \mathbf{w}_{t,k}, \kappa^{\mathrm{ems}}) \tag{26}$$

$$+ \sum_{t=2}^{T} \sum_{k=1}^{K} \log p(\mathbf{w}_{t,k} \mid \mathbf{w}_{t-1,k}, \kappa^{\mathrm{trans}}) \tag{27}$$

$$= \sum_{k=1}^{K} \log C_D(\kappa_{0,k}) + \kappa_{0,k} \boldsymbol{\mu}_{0,k}^{\top} \mathbf{w}_{1,k} \tag{28}$$

$$+ \sum_{t=1}^{T} \sum_{n=1}^{N_t} \sum_{k=1}^{K} c_{t,n,k} \big( \log \pi_{t,k} + \log C_D(\kappa^{\mathrm{ems}}) + \kappa^{\mathrm{ems}} \mathbf{w}_{t,k}^{\top} \mathbf{h}_{t,n} \big) \tag{29}$$

$$+ \sum_{t=2}^{T} \sum_{k=1}^{K} \log C_D(\kappa^{\mathrm{trans}}) + \kappa^{\mathrm{trans}} \mathbf{w}_{t-1,k}^{\top} \mathbf{w}_{t,k} \tag{30}$$

where $\kappa_{0,k}$ and $\boldsymbol{\mu}_{0,k}$ denote the prior parameters for $t = 1$. In practice, we set $\boldsymbol{\mu}_{0,k}$ to the source weights and $\kappa_{0,k} = 100$ (see App. C).

**Variational EM objective**   As described in Sec. 3.3, we approximate the posterior $p(\mathbf{W}_{1:T}, \mathbf{C}_{1:T} \mid \mathbf{H}_{1:T})$ with a variational distribution $q(\mathbf{W}_{1:T}, \mathbf{C}_{1:T})$ assuming the factorized form

$$q(\mathbf{W}_{1:T}, \mathbf{C}_{1:T}) = \prod_{t=1}^{T} \prod_{k=1}^{K} q(\mathbf{w}_{t,k}) \prod_{n=1}^{N_t} q(\mathbf{c}_{t,n}), \tag{31}$$

where we parameterize $q(\mathbf{w}_{t,k})$ and $q(\mathbf{c}_{t,n})$ with

$$q(\mathbf{w}_{t,k}) = \mathtt{vMF}(\cdot; \boldsymbol{\rho}_{t,k}, \gamma_{t,k}), \quad q(\mathbf{c}_{t,n}) = \mathtt{Cat}(\cdot; \boldsymbol{\lambda}_{t,n}), \quad \forall t, n, k. \tag{32}$$

We obtain the variational EM objective

$$\arg \max_{\phi} \mathbb{E}_q \big[ \log p(\mathbf{H}_{1:T}, \mathbf{W}_{1:T}, \mathbf{C}_{1:T}) \big], \tag{33}$$

where $\mathbb{E}_{q(\mathbf{W}_{1:T}, \mathbf{C}_{1:T})}$ is denoted $\mathbb{E}_q$ to reduce clutter.

**E-step**   Taking the expectation of the complete-data log likelihood (Eqn. (25)) with respect to the variational distribution (Eqn. (31)) gives

$$\mathbb{E}_q[\log p(\mathbf{H}_{1:T}, \mathbf{W}_{1:T}, \mathbf{C}_{1:T})] = \sum_{k=1}^{K} \log C_D(\kappa_{0,k}) + \kappa_{0,k} \boldsymbol{\mu}_{0,k}^{\top} \mathbb{E}_q[\mathbf{w}_{1,k}]$$

$$+ \sum_{t=1}^{T} \sum_{n=1}^{N_t} \sum_{k=1}^{K} \mathbb{E}_q[c_{t,n,k}] \big( \log \pi_{t,k} + \log C_D(\kappa^{\mathrm{ems}}) + \kappa^{\mathrm{ems}} \mathbb{E}_q[\mathbf{w}_{t,k}]^{\top} \mathbf{h}_{t,n} \big)$$

$$+ \sum_{t=2}^{T} \sum_{k=1}^{K} \log C_D(\kappa^{\mathrm{trans}}) + \kappa^{\mathrm{trans}} \mathbb{E}_q[\mathbf{w}_{t-1,k}]^{\top} \mathbb{E}_q[\mathbf{w}_{t,k}] \tag{34}$$

Solving for the variational parameters, we obtain

$$\lambda_{t,n,k} = \frac{\pi_{t,k} C_D(\kappa^{\text{ems}}) \exp\left\{\kappa^{\text{ems}} \mathbb{E}_q[\mathbf{w}_{t,k}]^\top \mathbf{h}_{t,n}\right\}}{\sum_{j=1}^K \pi_{t,j} C_D(\kappa^{\text{ems}}) \exp\left\{\kappa^{\text{ems}} \mathbb{E}_q[\mathbf{w}_{t,j}]^\top \mathbf{h}_{t,n}\right\}}, \tag{35}$$

$$\gamma_{t,k} = ||\beta_{t,k}||, \quad \boldsymbol{\rho}_{t,k} = \beta_{t,k}/\gamma_{t,k}, \tag{36}$$

$$\beta_{t,k} = \kappa^{\text{trans}} \mathbb{I}_{\{t>1\}} \mathbb{E}_q[\mathbf{w}_{t-1,k}] + \kappa^{\text{ems}} \sum_{n=1}^{N_t} \mathbb{E}_q[c_{t,n,k}]\mathbf{h}_{t,n} + \kappa^{\text{trans}} \mathbb{I}_{\{t<T\}} \mathbb{E}_q[\mathbf{w}_{t+1,k}], \tag{37}$$

The expectations are given by

$$\mathbb{E}_q[c_{t,n,k}] = \lambda_{t,n,k} \tag{38}$$

$$\mathbb{E}_q[\mathbf{w}_{t,k}] = A_D(\gamma_{t,k})\,\boldsymbol{\rho}_{t,k}, \tag{39}$$

where $A_D(\kappa) = \frac{I_{D/2}(\kappa)}{I_{D/2-1}(\kappa)}$ and $I_v(a)$ denotes the modified Bessel function of the first kind with order $v$ and argument $a$.

**M-step** Maximizing objective (Eqn. (33)) with respect to the model parameters $\phi = \{\kappa^{\text{trans}}, \kappa^{\text{ems}}, \boldsymbol{\pi}_{1:T}\}$ gives

$$\hat{\kappa}^{\text{trans}} = \frac{\bar{r}^{\text{trans}} D - (\bar{r}^{\text{trans}})^3}{1 - (\bar{r}^{\text{trans}})^2}, \quad \text{with} \quad \bar{r}^{\text{trans}} = \left\| \frac{\sum_{t=2}^T \sum_{k=1}^K \mathbb{E}_q[\mathbf{w}_{t-1,k}]^\top \mathbb{E}_q[\mathbf{w}_{t,k}]}{(T-1) \times K} \right\| \tag{40}$$

$$\hat{\kappa}^{\text{ems}} = \frac{\bar{r}^{\text{ems}} D - (\bar{r}^{\text{ems}})^3}{1 - (\bar{r}^{\text{ems}})^2}, \quad \text{with} \quad \bar{r}^{\text{ems}} = \left\| \frac{\sum_{t=1}^T \sum_{k=1}^K \sum_{n=1}^{N_t} \mathbb{E}_q[c_{t,n,k}]\,\mathbb{E}_q[\mathbf{w}_{t,k}]^\top \mathbf{h}_{t,n}}{\sum_{t=1}^T N_t} \right\| \tag{41}$$

$$\pi_{t,k} = \frac{\sum_{n=1}^{N_t} \mathbb{E}_q[c_{t,n,k}]}{N_t} \tag{42}$$

Here we made use of the approximation from Banerjee et al. (2005) to compute an estimate for $\kappa$,

$$\hat{\kappa} = \frac{\bar{r} D - \bar{r}^3}{1 - \bar{r}^2}, \quad \text{with} \quad \bar{r} = A_D(\hat{\kappa}). \tag{43}$$

---

**Algorithm 3** EM for STAD-vMF

---

1: Input: prototypes, cluster assignments, mixing coefficients, representations $\{\mathbf{W}_\tau, \mathbf{C}_\tau, \boldsymbol{\pi}_\tau, \mathbf{H}_\tau\}_{\tau \in S_t}$, transition and emission parameters $\kappa^{\text{trans}}, \kappa^{\text{ems}}$

2: **for** $\tau \in S_t$ **do**

3:      *E-Step*:

4:      Compute: $\lambda_{\tau,n,k} = \dfrac{\pi_{\tau,k} C_D(\kappa^{\text{ems}}) \exp\left\{\kappa^{\text{ems}} \mathbb{E}_q[\mathbf{w}_{\tau,k}]^\top \mathbf{h}_{\tau,n}\right\}}{\sum_{j=1}^{K} \pi_{\tau,j} C_D(\kappa^{\text{ems}}) \exp\left\{\kappa^{\text{ems}} \mathbb{E}_q[\mathbf{w}_{\tau,j}]^\top \mathbf{h}_{\tau,n}\right\}}$

5:      Get expectation of cluster assignments: $\mathbb{E}_q[c_{\tau,n,k}] = \lambda_{\tau,n,k}$

6:      Compute: $\beta_{\tau,k} = \kappa^{\text{trans}} \mathbb{I}_{\{\tau>1\}} \mathbb{E}_q[\mathbf{w}_{\tau-1,k}] + \kappa^{\text{ems}} \sum_{n=1}^{N_\tau} \mathbb{E}_q[c_{\tau,n,k}] \mathbf{h}_{\tau,n} + \kappa^{\text{trans}} \mathbb{I}_{\{\tau<T\}} \mathbb{E}_q[\mathbf{w}_{\tau+1,k}]$

7:      Compute: $\gamma_{\tau,k} = \|\beta_{\tau,k}\|$

8:      Compute: $\boldsymbol{\rho}_{\tau,k} = \beta_{\tau,k}/\gamma_{\tau,k}$

9:      Get expectation of prototypes: $\mathbb{E}_q[\mathbf{w}_{\tau,k}] = A_D(\gamma_{\tau,k})\boldsymbol{\rho}_{\tau,k}$

10:      *M-Step*:

11:      Compute: $\pi_{\tau,k} = \dfrac{\sum_{n=1}^{N_\tau} \mathbb{E}_q[c_{\tau,n,k}]}{N_\tau}$

12: **end for**

13: Compute: $\bar{r}^{\text{trans}} = \left\|\dfrac{\sum_{\tau \in S_t} \sum_{k=1}^{K} \mathbb{E}_q[\mathbf{w}_{\tau-1,k}]^\top \mathbb{E}_q[\mathbf{w}_{\tau,k}]}{(|S_t|-1) \times K}\right\|$

14: Compute: $\kappa^{\text{trans}} = \dfrac{\bar{r}^{\text{trans}} D - (\bar{r}^{\text{trans}})^3}{1 - (\bar{r}^{\text{trans}})^2}$

15: Compute: $\bar{r}^{\text{ems}} = \left\|\dfrac{\sum_{\tau \in S_t} \sum_{k=1}^{K} \sum_{n=1}^{N_\tau} \mathbb{E}_q[c_{\tau,n,k}] \mathbb{E}_q[\mathbf{w}_{\tau,k}]^\top \mathbf{h}_{\tau,n}}{\sum_{\tau \in S_t} N_\tau}\right\|$

16: Compute: $\kappa^{\text{ems}} = \dfrac{\bar{r}^{\text{ems}} D - (\bar{r}^{\text{ems}})^3}{1 - (\bar{r}^{\text{ems}})^2}$

17: Assign: $\mathbf{W}_t = (\boldsymbol{\rho}_{t,1}, \ldots, \boldsymbol{\rho}_{t,K})$

18: Assign: $\mathbf{C}_t = (\boldsymbol{\lambda}_{t,1}, \ldots, \boldsymbol{\lambda}_{t,N_t})$

19: **return** $\mathbf{W}_t, \mathbf{C}_t, \boldsymbol{\pi}_t, \kappa^{\text{trans}}, \kappa^{\text{ems}}$

---

# C  Experimental Details

We next list details on the experimental setup and hyperparameter configurations. All experiments are performed on NVIDIA RTX 6000 Ada with 48GB memory.

## C.1  Datasets

- **Yearbook** (Ginosar et al., 2015): a dataset of portraits of American high school students taken across eight decades. Data shift in the students' visual appearance is introduced by changing beauty standards, group norms, and demographic changes. We use the Wild-Time (Yao et al., 2022) pre-processing and evaluation procedure resulting into 33,431 images from 1930 to 2013. Each $32 \times 32$ pixel, grey-scaled image is associated with the student's gender as a binary target label. Images from 1930 to 1969 are used for training; the years 1970 - 2013 for testing.
- **EVIS**: the *evolving image search* (EVIS) dataset (Zhou et al., 2022b) consists of images of 10 electronic product and vehicle categories retrieved from Google search, indexed by upload date. The dataset captures shift caused by rapid technological advancements, leading to evolving designs across time. It includes 57,600 RGB images of 256x256 pixels from 2009 to 2020. Models are trained on images from 2009-2011 and evaluated on images from 2012-2020.
- **FMoW-Time**: the *functional map of the world* (FMoW) dataset (Koh et al., 2021) maps $224 \times 224$ RGB satellite images to one of 62 land-use categories. Distribution shift is introduced by technical advancement and economic growth changing how humans make use of the land. FMoW-Time (Yao et al., 2022) is an adaptation of FMoW-WILDS (Koh et al., 2021; Christie et al., 2018), splitting 141,696 images into a training period (2002-2012) and a testing period (2013-2017).
- **CIFAR-10.1** (Recht et al., 2019): a reproduction of CIFAR-10 (Krizhevsky et al., 2009) assembled from the same data source by following the same cleaning procedure. The dataset contains 2,000 $32 \times 32$ pixel images of 10 classes. Models are trained on the original CIFAR-10 train set.
- **ImageNetV2** (Recht et al., 2019): a reproduction of ImageNet (Deng et al., 2009) with 10,000 images of 1,000 classes scaled to $224 \times 224$ pixels. Models are trained on the original ImageNet.
- **CIFAR-10-C**: a dataset derived from CIFAR-10, to which 15 corruption types are applied with 5 severity levels (Hendrycks & Dietterich, 2019). We mimic a gradual distribution shift by increasing the corruption severity starting from the lowest level (severity 1) to the most sever corruption (severity 5). This results in a test stream of $5 \times 10,000$ images per corruption type.

## C.2  Source Architectures

- **CNN**: We employ the four-block convolutional neural network trained by Yao et al. (2022) to perform the binary gender prediction on the yearbook dataset. Presented results are averages over three different seeds trained with empirical risk minimization. The dimension of the latent representation space is 32.
- **WideResNet**: For the CIFAR-10 experiments, we follow Song et al. (2023); Wang et al. (2021) and use the pre-trained WideResNet-28 (Zagoruyko & Komodakis, 2016) model from the RobustBench benchmark (Croce et al., 2021). The latent representation have a dimension of 512.
- **DenseNet**: For FMoW-Time, we follow the backbone choice of Koh et al. (2021); Yao et al. (2022) and use DenseNet121 (Huang et al., 2017) for the land use classification task. Weights for three random trainings seeds are provided by Yao et al. (2022). We use the checkpoints for plain empirical risk minimization. The latent representation dimension is 1024.
- **ResNet**: For EVIS, we follow Zhou et al. (2022b) and use their ResNet-18 (He et al., 2016) model with a representation dimension of 512 and train on three random seeds. For ImageNet-V2, we follow Song et al. (2023) and employ the standard pre-trained ResNet-50 model from RobustBench (Croce et al., 2021). Latent representations are of dimension 2048.
- **ViT-base**: For ImageNet-C, we employ ViT-base (Dosovitskiy et al., 2021) from the timm library (Wightman, 2019). Latent representations are of dimension 768.

## C.3 Baselines

- **Source Model**: the un-adapted original model.
- **BatchNorm (BN) Adaptation** (Schneider et al., 2020; Nado et al., 2020): aims to adapt the source model to distributions shift by collecting normalization statistics (mean and variance) of the test data.
- **Test Entropy Minimization (TENT)** (Wang et al., 2021): goes one step further and optimizes the BN transformation parameters (scale and shift) by minimizing entropy on test predictions.
- **Continual Test-Time Adaptation (CoTTA)** (Wang et al., 2022b): optimizes all model parameters with an entropy objective on augmentation averaged predictions and combines it with stochastic weight restore to prevent catastrophic forgetting.
- **Source HypOthesis Transfer (SHOT)** (Liang et al., 2020) adapts the feature extractor via an information maximization loss in order to align the representations with the source classifier.
- **Sharpness-Aware Reliable Entropy Minimization (SAR)** (Niu et al., 2023) filters out samples with large gradients based on their entropy values and encourages convergence to a flat minimum.
- **Robust Test-time Adaptation (RoTTA)** (Yuan et al., 2023) proposes a robust BN layer and the use of a class-balanced memory bank to address simultaneous covariate and label shift.
- **Laplacian Adjusted Maximum likelihood Estimation (LAME)** (Boudiaf et al., 2022) regularizes the likelihood of the source model with a Laplacian correction term that encourages neighbouring representations to be assigned to the same class.
- **Test-Time Template Adjuster (T3A)** (Iwasawa & Matsuo, 2021) computes new class prototypes by a running average of low entropy representations.

## C.4 Implementation Details and Hyperparameters

By the nature of test-time adaptation, choosing hyperparameters is tricky (Zhao et al., 2023b) since one cannot assume access to a validation set of the test distribution in practise. To ensure we report the optimal performance on new or barely used datasets (Yearbook, EVIS, FMoW, CIFAR-10.1 and ImageNetV2), we perform a grid search over hyperparameters as suggested in the original papers. We perform separate grid searches for the uniform label distribution and online imbalanced label distribution setting. Reported performance correspond to the best setting. If the baselines were studied in the gradual CIFAR-10-C setting by Wang et al. (2022b), we use their hyperparameter setup; otherwise, we conduct a grid search as described earlier. Unless there is a built-in reset (SAR) or convergence criteria (LAME) all methods run without reset and one optimization step is performed. We use the same batch sizes for all baselines. For Yearbook we comprise all samples of a year in one batch resulting in a batch size of 2048. To create online class imbalance, we reduce the batch size to 64. We use a batch size of 100 for EVIS, CIFAR.10.1 and CIFAR-10-C and 64 for FMoW-Time and ImageNetV2.

**BN** (Schneider et al., 2020; Nado et al., 2020) Normalization statistics during test-time adaptation are a running estimates of both the training data and the incoming test statistics. No hyperparameter optimization is necessary here.

**TENT** (Wang et al., 2021) Like in BN, the normalization statistics are based on both training and test set. As in Wang et al. (2021), we use the same optimizer settings for test-time adaptation as used for training, except for the learning rate that we find via grid search on $\{1e^{-3}, 1e^{-4}, 1e^{-5}, 1e^{-6}, 1e^{-7}\}$. Adam optimizer (Kingma & Ba, 2015) is used. For CIFAR-10-C, we follow the hyperparameter setup of Wang et al. (2022b) and use Adam optimizer with learning rate $1e-3$.

**CoTTA** (Wang et al., 2022b) We use the same optimizer as used during training (Adam optimizer Kingma & Ba (2015)). For hyperparameter optimization we follow the parameter suggestions by Wang et al. (2022b) and conduct a grid search for the learning rate ($\{1e^{-3}, 1e^{-4}, 1e^{-5}, 1e^{-6}, 1e^{-7}\}$), EMA factor ($\{0.99, 0.999, 0.9999\}$) and restoration factor ($\{0, 0.001, 0.01, 0.1\}$). Following Wang et al. (2022b), we determine the augmentation confidence threshold by the 5% percentile of the softmax prediction confidence from the source model on the source images. For the well-studied CIFAR-10-C dataset, we follow the setting of Wang et al. (2022b) and

use Adam optimizer with learning rate $1e - 3$. The EMA factor is set to 0.999, the restoration factor is 0.01 and the augmentation confidence threshold is 0.92.

**SHOT** (Liang et al., 2020) We perform a grid search for the learning rate over $\{1e^{-3}, 1e^{-4}\}$ and for $\beta$, the scaling factor for the loss terms, over $\{0.1, 0.3\}$.

**SAR** (Niu et al., 2023) We conduct a grid search over the learning rate selecting among $\{1e^{-2}, 1e^{-3}, 1e^{-4}, 0.00025\}$. Like the authors, we compute the $E_0$ threshold as a function of number of classes $0.4 \times \ln K$, use SDG, a moving average factor of 0.9, and the reset threshold of 0.2. While (Niu et al., 2023) apply SAR only to models with layer or group normalization and update those layers, we also evaluate SAR on source architectures with batch normalization, following prior work (Zhao et al., 2023b). As noted in (Niu et al., 2023), BN layers are less effective for small batch sizes, which accounts for the reduced performance of SAR in this setting (Fig. 4 and Tab. 10).

**RoTTA** (Yuan et al., 2023) We conduct a search over the learning rate among $\{1e^{-3}, 1e^{-4}, 1e^{-5}\}$. As in the experiments of the original paper, we use the default values for $\lambda_t = 1.0$, $\lambda_u = 1.0$, $\alpha = 0.05$ and $\nu = 0.001$.

**LAME** (Boudiaf et al., 2022) The only hyperparameter is the choice of affinity matrix. Like Boudiaf et al. (2022) we use a $k$-NN affinity matrix and select the number of nearest neighbours among $\{1, 3, 5\}$.

**T3A** (Iwasawa & Matsuo, 2021) We test different values for the hyperparameter $M$. The $M$-th largest entropy values are included in the support set used for computing new prototypes. We test the values $\{1, 5, 20, 50, 100, \texttt{None}\}$. $\texttt{None}$ corresponds to no threshold, i.e. all samples are part of the support set.

**STAD-vMF** The hyperparameters are the initialization values of the transition concentration parameter $\kappa^{trans}$, emission concentration parameter $\kappa^{ems}$ and the sliding window size $s$. We chose the concentration parameters from $\{100, 1000\}$. Tab. 7 lists employed settings. We use a default window size of $s = 3$. For Yearbook, we employ class specific noise parameters $\kappa_k^{\mathrm{trans}}$ and $\kappa_k^{\mathrm{ems}}$ as discussed in Sec. 3.3. For the other datasets, we found a more restricted noise model beneficial. Particularly, we use global concentration parameters, $\kappa^{\mathrm{trans}}$ and $\kappa^{\mathrm{ems}}$, and follow suggestions by Gopal & Yang (2014) to keep noise concentration parameters fixed instead of learning them via maximum likelihood (line 13 - 16 in Algorithm 3). Keeping them fix acts as a regularization term as it controls the size of the cluster (via $\kappa^{\mathrm{ems}}$) and the movement of the prototypes (via $\kappa^{\mathrm{trans}}$). Low concentration values generally correspond to more adaptation flexibility while larger values results in a more conservative and rigid model.

**STAD-Gauss** We initialize the mixing coefficients with $\pi_{t,k} = \frac{1}{K} \forall t, k$, the transition covariance matrix with $\boldsymbol{\Sigma}^{\mathrm{trans}} = 0.01 \times \mathbf{I}$ and the emission covariance matrix with $\boldsymbol{\Sigma}^{\mathrm{ems}} = 0.5 \times \mathbf{I}$. We found a normalization of the representations to be also beneficial for STAD-Gauss. Note that despite normalization, the two models are not equivalent. STAD-Gauss models the correlation between different dimensions of the representations and is therefore more expressive, while STAD-vMF assumes an isotropic variance.

Table 7: Hyperparameters employed for STAD-vMF

| Dataset | $\kappa^{trans}$ | $\kappa^{ems}$ |
|---|---|---|
| Yearbook (covariate shift) | 100 | 100 |
| Yearbook (+ label shift) | 1000 | 100 |
| EVIS (covariate shift) | 1000 | 1000 |
| EVIS (+ label shift) | 1000 | 100 |
| FMoW-Time (covariate shift) | 100 | 100 |
| FMoW-Time (+ label shift) | 1000 | 100 |
| CIFAR-10.1 (covariate shift) | 1000 | 1000 |
| CIFAR-10.1 (+ label shift) | 1000 | 100 |
| ImageNetV2 (covariate shift) | 100 | 1000 |
| ImageNetV2 (+ label shift) | 1000 | 100 |
| CIFAR-10-C (covariate shift) | 1000 | 100 |

# D   Additional Results

## D.1   Domain Adaptation Benchmarks

To study the limitations and applicability of our method STAD, we also test adaptation performance on non-gradual shifts. For that, we use the domain adaptation benchmark PACS, which comprises images of 10 classes across four categorical domains (*photo*, *art-painting*, *cartoon* and *sketch*). We use DomainBed (Gulrajani & Lopez-Paz, 2021) to train a ResNet-50 model with BN. We test two settings. In the first setting, we follow Iwasawa & Matsuo (2021); Jang et al. (2023) and train the model on three domains and adapt it on the held-out domain. In the second setting, we follow Gui et al. (2024) and train the model on the *photo* domain and adapt it on the remaining domains sequentially. The second setting is more difficult as the model is exposed to only a single domain during training and needs to leverage several abrupt changes in the distribution over a longer test stream.

Tab. 8 shows adaptation performance for the first setting. Adaptation gains are generally smaller than on corruption datasets, with TENT, RoTTA, and SAR improving most upon the source model by 2–3 percentage points. While STAD achieves a more modest improvement of 1 percentage point, it remains the best-performing method among classifier adaptation approaches.

Table 8: Accuracy on **domain adaptation benchmarks** under covariate shift and uniform label distribution. The source model is trained on three domains and tested on the remaining one, rotating the test domain for averaging.

| Method | PACS |
|---|---|
| Source model | $82.99 \pm 8.87$ |
| *adapt feature extractor* | |
| BN | $82.85 \pm 9.57$ |
| TENT | $\mathbf{85.30} \pm 7.33$ |
| CoTTA | $83.59 \pm 8.46$ |
| SHOT | $83.30 \pm 9.01$ |
| SAR | $85.03 \pm 7.71$ |
| RoTTA | $\underline{85.11} \pm 7.73$ |
| *adapt classifier* | |
| LAME | $83.31 \pm 8.90$ |
| T3A | $83.68 \pm 9.14$ |
| **STAD-vMF** (ours) | $83.91 \pm 8.58$ |

Results for the second setting are shown in Tab. 9. Consistent with Gui et al. (2024), we observe a decreasing performance across all TTA methods over the course of adaptation, highlighting the challenge posed by multiple non-gradual domain shifts. Nevertheless, all TTA methods, except LAME, improve upon the source model by over 10 percentage points on average. Despite the highly non-gradual nature of this test setting, STAD-vMF performs comparably to the baselines, achieving the third-best performance overall. These findings strengthens the results in Sec. 5.2, which suggest that STAD is applicable beyond gradual, temporal distribution shifts similar as other TTA methods.

Table 9: Accuracy on **domain adaptation benchmarks** under covariate shift and uniform label distribution. The source model is trained on the photo domain and TTA methods adapt to the remaining domains sequentially. Results show average over three random training seeds. N/A indicates that adaptation is not applied to the source domain.

| Method | P | $\rightarrow$ A | $\rightarrow$ C | $\rightarrow$ S | Mean |
|---|---|---|---|---|---|
| Source | $99.34 \pm 0.57$ | $63.10 \pm 1.55$ | $38.37 \pm 4.99$ | $41.51 \pm 2.99$ | $47.66 \pm 1.29$ |
| *adapt feature extractor* | | | | | |
| BN | N/A | $68.03 \pm 1.98$ | $61.15 \pm 0.34$ | $49.64 \pm 0.28$ | $59.61 \pm 0.52$ |
| TENT | N/A | $68.05 \pm 2.11$ | $61.53 \pm 0.52$ | $51.33 \pm 1.35$ | $60.30 \pm 0.28$ |
| CoTTA | N/A | $63.82 \pm 3.23$ | $59.36 \pm 1.35$ | $56.74 \pm 2.70$ | $59.97 \pm 2.17$ |
| SHOT | N/A | $67.91 \pm 2.04$ | $\mathbf{63.17} \pm 0.86$ | $\mathbf{57.90} \pm 1.22$ | $\mathbf{62.99} \pm 0.58$ |
| SAR | N/A | $68.34 \pm 1.81$ | $61.49 \pm 0.36$ | $52.06 \pm 1.10$ | $60.63 \pm 0.67$ |
| RoTTA | N/A | $\mathbf{68.73} \pm 1.13$ | $58.52 \pm 1.32$ | $52.43 \pm 1.19$ | $59.89 \pm 0.19$ |
| *adapt classifier* | | | | | |
| LAME | N/A | $62.73 \pm 1.72$ | $37.80 \pm 5.09$ | $41.01 \pm 2.90$ | $47.18 \pm 1.26$ |
| T3A | N/A | $68.28 \pm 1.55$ | $\underline{62.05} \pm 0.67$ | $\underline{54.80} \pm 0.97$ | $\underline{61.71} \pm 0.40$ |
| **STAD-vMF** (ours) | N/A | $\underline{68.59} \pm 2.51$ | $61.65 \pm 0.39$ | $52.16 \pm 0.49$ | $60.80 \pm 0.72$ |

## D.2 Single Sample Adaptation

Table 10: Adaptation accuracy on temporal distribution shift with single sample adaptation (batch size 1) for both covariate shift and additional online label shift: Table shows values as plotted in Fig. 4. Most methods collapse when only provided with one sample per adaptation step. STAD can improve upon the source model in 4 out of 6 scenarios.

| Model | Yearbook covariate shift | + label shift | EVIS covariate shift | + label shift | FMoW covariate shift | + label shift |
|---|---|---|---|---|---|---|
| Source | 81.30 ± 4.18 | | 56.59 ± 0.92 | | 68.94 ± 0.20 | |
| *adapt feature extractor* | | | | | | |
| BN | 73.32 ± 6.90 | 73.32 ± 6.90 | 11.12 ± 0.97 | 11.12 ± 0.97 | 3.46 ± 0.03 | 3.46 ± 0.03 |
| TENT | 61.33 ± 9.42 | 61.45 ± 9.46 | 10.80 ± 0.84 | 10.78 ± 0.81 | 4.00 ± 0.87 | 3.99 ± 0.87 |
| CoTTA | 55.91 ± 5.26 | 56.71 ± 6.12 | 10.04 ± 0.08 | 9.88 ± 0.34 | 3.42 ± 0.14 | 3.42 ± 0.03 |
| SHOT | 53.71 ± 3.77 | 51.48 ± 1.68 | 20.23 ± 1.40 | 16.36 ± 1.67 | 3.84 ± 0.21 | 4.01 ± 0.44 |
| SAR | 73.32 ± 6.89 | 73.38 ± 7.01 | 11.12 ± 0.97 | 11.12 ± 0.97 | 3.46 ± 0.03 | 3.46 ± 0.03 |
| *adapt classifier* | | | | | | |
| T3A | 83.51 ± 2.54 | 83.44 ± 2.60 | 57.63 ± 0.77 | 57.40 ± 0.76 | 66.78 ± 0.24 | 66.87 ± 0.27 |
| LAME | 81.29 ± 4.18 | 81.30 ± 4.18 | 56.59 ± 0.91 | 56.59 ± 0.91 | 68.94 ± 0.20 | 68.94 ± 0.20 |
| **STAD-vMF** | 84.32 ± 2.03 | 81.49 ± 4.23 | 56.15 ± 0.98 | 58.02 ± 0.77 | 68.88 ± 0.29 | 71.22 ± 0.40 |

## D.3 STAD in Combination with BN

Table 11: We explored whether STAD, which adapts the classifier, can be effectively combined with TTA methods like BN adaptation, which targets the feature extractor. The results are mixed. On the covariate shift of Yearbook, combining the two methods improves performance beyond what each achieves individually. However, on other datasets, the combination generally results in decreased performance.

| Model | Yearbook covariate shift | + label shift | EVIS covariate shift | + label shift | FMoW covariate shift | + label shift |
|---|---|---|---|---|---|---|
| Source | 81.30 ± 4.18 | | 56.59 ± 0.92 | | 68.94 ± 0.20 | |
| BN | 84.54 ± 2.10 | 70.47 ± 0.33 | 45.72 ± 2.79 | 14.48 ± 1.02 | 67.60 ± 0.44 | 10.14 ± 0.04 |
| STAD-vMF | 85.50 ± 1.34 | 84.46 ± 1.19 | 56.67 ± 0.82 | 62.08 ± 1.11 | 68.87 ± 0.06 | 86.25 ± 1.18 |
| STAD-vMF + BN | 86.20 ± 1.23 | 69.96 ± 0.39 | 44.23 ± 2.88 | 15.18 ± 1.68 | 66.97 ± 0.46 | 9.26 ± 1.97 |
| STAD-Gauss | 86.22 ± 0.84 | 84.67 ± 1.46 | – | – | – | – |
| STAD-Gauss + BN | 86.56 ± 1.08 | 70.12 ± 0.33 | – | – | – | – |

### D.4 ImageNet-C

Table 12: Accuracy on gradually increasing corruptions of ImageNet-C using ViT-Base as the source model. BN and RoTTA are specific to batch normalization layers and are thus ineffective on ViT; they are omitted from this experiment. As discussed in Sec. 5.2, methods that adapt the feature extractor achieve greater adaptation gains on synthetic corruptions—except for CoTTA, which collapses. On this dataset, STAD provides only marginal improvement over the source model.

| Method | Corruption severity | | | | | Mean |
| --- | --- | --- | --- | --- | --- | --- |
| | 1 | 2 | 3 | 4 | 5 | |
| Source | 66.66 | 59.71 | 53.88 | 43.97 | 32.43 | 51.33 |
| *adapt feature extractor* | | | | | | |
| TENT | 68.41 | 66.01 | 63.85 | 59.18 | 52.89 | **62.07** |
| CoTTA | 48.02 | 34.87 | 28.12 | 19.78 | 12.71 | 28.70 |
| SHOT | 67.51 | 64.41 | 62.33 | 57.52 | 51.19 | 60.59 |
| SAR | 67.69 | 64.70 | 62.33 | 57.55 | 51.01 | 60.66 |
| *adapt classifier* | | | | | | |
| LAME | 66.48 | 59.52 | 53.64 | 43.76 | 32.21 | 51.12 |
| T3A | 66.71 | 61.31 | 56.59 | 46.75 | 34.22 | 53.12 |
| **STAD-vMF** (ours) | 66.60 | 59.81 | 54.05 | 44.25 | 32.80 | 51.50 |

### D.5 Comparison to Supervised Oracle

We investigate how far STAD, which operates unsupervised and does not require labels, can close the gap to a supervised approach that makes use of labels. Obviously, a supervised approach is superior, as it can directly learn the function mapping input samples to target labels. In contrast, TTA methods rely solely on signals from the input and therefore have strictly less information available. To evaluate how far this gap can be bridged, we continuously fine-tune the source model at each timestep using a small portion of labeled samples. For this, we use another held-out set from the Wild-Time pipeline, which is 10% the size of the adaptation test stream. At each timestep, we fine-tune the model for one epoch on this split and then evaluate the performance on the regular test set.

We find that the supervised model achieves an average accuracy of 90.67% over the entire test stream. Comparing this to the source model at 81.30%, STAD-Gauss at 86.22%, and STAD-vMF at 85.50% (see Tab. 2), we observe that STAD can partially close the gap between the unadapted source model and the fine-tuned model. Fig. 7 further reveals that STAD-vMF is on par with the fine-tuned classifier at certain time steps (1986, 1995, 1997). Additionally, we observe that the severity of the temporal distribution shift hampers also the supervised model's ability to regain in-distribution accuracy. For instance, in the 1970s and 1980s, the performance of the supervised model is 20 points lower than its in-distribution accuracy (nearly 100%).

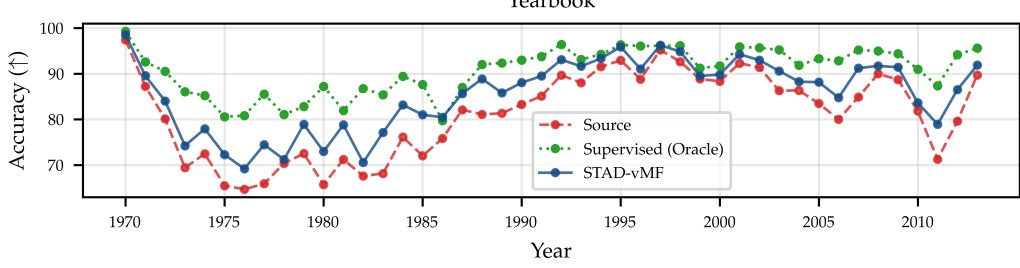

Figure 7: Accuracy over time for the temporal distribution shift on Yearbook averaged over three random training seeds.

## D.6 Cluster Visualization

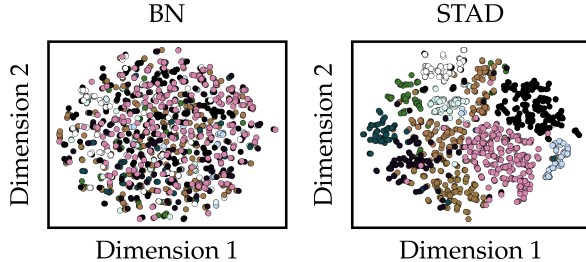

Figure 8: t-SNE visualization of the representation space of FMoW-Time (year 2013) under joint covariate and label shift: We visualize the cluster structure in representation space. Colors indicate ground truth class labels for the 10 most common classes. Adapting with BN destroys the cluster structure, resulting in inseparable clusters. In contrast, STAD operates on linearly separable representations.

## D.7 Computational Complexity and Runtime

The complexity of STAD-vMF scales linearly with the sliding window size $s$, batch size $N_t$, representation dimension $D$ and number of classes $K$, i.e. $\mathcal{O}(s \times N_t \times D \times K)$. Note that the sliding window size $s$ is fixed at $s = 3$ and operations over the batch size, number of classes and representation dimension can be fully parallelized.

Tab. 13 reports relative runtime compared to the source model for STAD and baselines. As results on ImageNet-V2 show, even for high dimensions ($D = 2048$) and large number of classes ($K = 1000$) STAD's runtime is comparable to baseline TTA methods.

Table 13: Relative runtime per batch compared to the source model

| Methods | Yearbook | FMoW-Time | ImageNetV2 |
|---|---|---|---|
| Source Model | 1.0 | 1.0 | 1.0 |
| BN | 1.0 | 1.1 | 1.1 |
| TENT | 1.4 | 6.4 | 7.2 |
| CoTTA | 17.1 | 200.0 | 327.3 |
| SHOT | 1.3 | 6.3 | 6.7 |
| SAR | 1.5 | 7.1 | 10.0 |
| RoTTA | 90.0 | 30.0 | 41.8 |
| LAME | 1.2 | 2.9 | 5.5 |
| T3A | 1.1 | 1.8 | 10.9 |
| STAD-vMF | 2.5 | 8.8 | 23.6 |
| STAD-Gauss | 3.3 | - | - |

## D.8 Sensitivity to Hyperparameters

In this section, we conduct a sensitivity analysis of the hyperparameters involved in STAD. We analyze sensitivity on two datasets: the temporal shift dataset Yearbook and the commonly used corruption benchmark CIFAR-10-C. All experiments are conducted under a uniform label distribution. When testing the sensitivity to a specific hyperparameter, all other hyperparameters are fixed at their default values (see App. C). Results show the average over three random training seeds for Yearbook and the average over 15 corruption types for CIFAR-10-C.

**Sensitivity to sliding window size** $s$    The window size determines the number of past time steps considered by the dynamic model. A small $s$ limits the influence of past prototypes, whereas a larger $s$ extends the considered history, giving more weight to past prototypes. However, large values of $s$ come at the cost of increased computational burden, as runtime scales linearly with window size. Tab. 14 suggests that increasing the window size could improve adaptation performance.

Table 14: Accuracy of STAD for different values of $s$

| $s$ | 3 | 5 | 7 |
|---|---|---|---|
| Yearbook | $85.4975 \pm 1.34$ | $85.5022 \pm 1.30$ | $85.5029 \pm 1.31$ |
| CIFAR-10-C | $76.9683 \pm 11.25$ | $76.9735 \pm 11.25$ | $76.9823 \pm 11.24$ |

**Sensitivity to $\kappa^{trans}$**    The transition concentration parameter $\kappa^{trans}$ regulates the transition noise and determines how far cluster prototypes move between different time steps. A high concentration value $\kappa^{trans}$ implies little movement of class prototypes, whereas low $\kappa^{trans}$ allows prototypes to move more. This parameter thus acts as a regularization factor between a more static and a more dynamic model. Tab. 15 displays the results. Performance changes only marginally for different values of the concentration parameter.

Table 15: Accuracy of STAD-vMF for different values of $\kappa^{\text{trans}}$

| $\kappa^{\text{trans}}$ | 50 | 100 | 500 | 1000 | 5000 |
|---|---|---|---|---|---|
| Yearbook | $85.5034 \pm 1.3031$ | $85.5034 \pm 1.3031$ | $85.5034 \pm 1.3031$ | $85.5034 \pm 1.3031$ | $85.4980 \pm 1.3099$ |
| CIFAR-10-C | $76.9684 \pm 11.2540$ | $76.9683 \pm 11.2543$ | $76.9685 \pm 11.2538$ | $76.9685 \pm 11.2538$ | $76.9688 \pm 11.2552$ |

**Sensitivity to $\kappa^{ems}$**    The emission concentration parameter $\kappa^{ems}$ regulates the emission noise and determines the spread of clusters. A high concentration value $\kappa^{ems}$ implies small, compact clusters, while low $\kappa^{ems}$ allows for widespread clusters. Results are shown in Tab. 16.

Table 16: Accuracy of STAD-vMF for different values of $\kappa^{\text{ems}}$

| $\kappa^{\text{ems}}$ | 50 | 100 | 500 | 1000 | 5000 |
|---|---|---|---|---|---|
| Yearbook | $85.5022 \pm 1.3036$ | $85.5034 \pm 1.3031$ | $85.5043 \pm 1.3019$ | $85.5058 \pm 1.3001$ | $85.5058 \pm 1.3001$ |
| CIFAR-10-C | $76.9679 \pm 11.2543$ | $76.9683 \pm 11.2543$ | $76.9689 \pm 11.2529$ | $76.9700 \pm 11.2514$ | $76.9700 \pm 11.2513$ |

