# OpenReview forum: "Temporal Test-Time Adaptation with State-Space Models"
_TMLR — Accepted by TMLR_

### Review · Reviewer_F8Aa · 2025-07-18

**Summary Of Contributions:**

This paper propose a new approach for temporal test-time adaptation, where distribution shifts change over time during inference. The proposed approach use a probabilistic state-space model to adapt the classifier by learning the classification features in the last layer. The authors evaluated their proposed approach on real-world temporal distribution shifts and demonstrate the effectiveness of proposed approach.

**Audience:**

Yes

**Broader Impact Concerns:**

None.

**Claims And Evidence:**

Yes

**Requested Changes:**

1. Could the authors discuss in detail the distribution shifts that the proposed approach can effectively handle?
2. Could the authors add more description on the motivation for Section 3.3? it is hard to understand this part.
3. Could you add a part that formally describe the proposed algorithm?

**Strengths And Weaknesses:**

Strengths:
1. This paper is well-written, with detailed approach description and experimental evaluation.
2. This problem this paper studies is important, as real-world distribution shifts are not static but could vary over time.


Weaknesses
1. I am a little unsure about the use of state-space models as framing the approach in order to avoid confusion with other popular deep learning architectures, e.g., Mamba. When I was reading the abstract and introduction, i was getting the sense that this paper is adapting Mamba models. I think a more accurate statement is that you are using probabilistic models to model the test-time distribution shift.
2. The paper lacks discussion on the distribution shifts you are tackling and the limitations, for instance what type of distribution shift the proposed approach would work less effectively.

---

> ### Author Response · Authors · 2025-08-12
>
> We thank you for your feedback and reviewing efforts. We address your points below.
>
> > *I am a little unsure about the use of state-space models as framing [...].*
>
> We understand that, given the recent prominence of structured state-space model architectures such as Mamba, the term “state-space models” may be associated with these approaches. However, probabilistic state-space models have been established in the literature for decades, and the terminology is well grounded in that tradition.
>
> **To clarify that our work builds on probabilistic rather than structured architectural SSMs, we have made the following changes to the paper:**
>
> Abstract:
>
> - *To address this, we propose STAD, a probabilistic state-space model that adapts a deployed model to temporal distribution shifts […]:* probabilistic state-space model → Bayesian filtering method
>
> Introduction:
>
> - *Specifically, we employ an SSM to track the evolution of the weight vectors in the final layer, where each vector represents a class, as distribution shift occurs […]:* SSM → probabilistic SSM based on Bayesian filtering
>
> Thank you for this comment, which has improved the presentation of our work.
>
> > *Could the authors discuss in detail the distribution shifts that the proposed approach can effectively handle?*
>
> We found that STAD is most effective under the following conditions:
>
> 1. **Distribution shift impacts last layer:** As studied by prior work [1], different types of distribution shifts are mitigated best in different network layers. Since STAD only adapts the last layer, it is less effective when the shift primarily affects earlier layers, which [1] finds to be the case for synthetic corruptions. We also observe that STAD is less effective on the corruption shifts of CIFAR-10-C (Table 4) and ImageNet-C (Table 12) than TTA methods adapting the feature extractor.
> 2. **Gradual change:** By design, STAD assumes that last-layer representations are temporally correlated. Abrupt changes that are not tractable for an SSM are thus challenging to mitigate. However, we did not find a scenario in practice where a sudden shift completely altered last-layer representations to the extent that STAD could not track it. For example, STAD handles sudden domain shifts on the PACS dataset competitively with other TTA methods (Tables 8 and 9).
> 3. **Label shift or covariate shift:** As stated in Section 2, STAD only leverages unlabeled test inputs and adapts to label shift and covariate shift. Concept drift that affects the conditional distribution $\mathbb{Q}\_t(\text{y} \mid \mathbf{x})$ cannot be mitigated without test labels.
>
> **We have added this discussion to Section 6 of the revised manuscript.**
>
> > *Could the authors add more description on the motivation for Section 3.3?*
>
> We assume you are asking for the motivation behind the von Mises–Fisher (vMF) variant described in Section 3.3. If we have misunderstood your point, please let us know.
>
> We provide reasons for the development of STAD-vMF in Section 3.2 (*Gaussian Model*) and Section 3.3 (first and last paragraphs). STAD-vMF is motivated by three key observations:
>
> (i) Prior work on distribution shifts has shown that the scale of representations and prototypes encodes biases of the source distribution. In particular, under a class-imbalanced source distribution $\mathbb{P}(\mathbf{x},\text{y})$, the learned prototypes of the dominant class will have a larger scale. Discarding the scale by normalizing representations and prototypes corresponds to discarding such biases specific to $\mathbb{P}(\mathbf{x},\text{y})$. STAD-vMF does so by operating on normalized representations and prototypes to better adapt to samples from $\mathbb{Q}_t(\mathbf{x})$ (see the first paragraph of Section 3.3).
> (ii) The Gaussian variant (Section 3.2) learns a full noise covariance matrix for each class, which causes runtime and memory usage to scale quadratically with the representation dimension $D$. This is prohibitive for high-dimensional features. STAD-vMF addresses this by learning a single scalar noise parameter per class, reducing both memory footprint and computation time.
> (iii) We show that the Softmax predictive distribution is theoretically equivalent to the posterior distribution over cluster assignments in STAD-vMF (Equation 10). This links STAD-vMF’s predictive distribution to standard classifier inference.
>
> > *Could you add a part that formally describe the proposed algorithm?*
>
> In the original submission, we formally described STAD-vMF in Algorithm 1 and Appendix B.2. During the rebuttal, **we have (i) moved Algorithm 1 to the main paper (Section 3.2) and generalized it to summarize both STAD-Gauss and STAD-vMF, and (ii) added Algorithms 2 and 3, which provide the specifics of the EM-step for the parametric choices of STAD-Gauss and STAD-vMF, respectively.**
>
> If there are any remaining aspects that you think would benefit from further clarification, please let us know, and we will be happy to address them.

---

### Review · Reviewer_q6BP · 2025-07-22

**Summary Of Contributions:**

The submission makes three key contributions to the field of test-time adaptation (TTA):

1. **Formalization of Temporal Test-Time Adaptation (TempTTA)**: It explicitly defines and details the setting of TempTTA, which focuses on handling gradually evolving distribution shifts over time—common in real-world scenarios but underexplored in TTA literature. The work demonstrates that such shifts pose significant challenges to existing TTA methods.

2. **Proposal of STAD**: A novel TempTTA method, State-space Test-time Adaptation (STAD), is introduced. STAD leverages probabilistic state-space models (SSMs) to model the dynamics of temporal distribution shifts in the representation space, adapting the model's final layer by tracking time-varying class prototypes. Unlike prior work, it explicitly models these dynamics, which is shown to be critical via ablation studies.

3. **Comprehensive Evaluation**: STAD is rigorously evaluated against prominent TTA baselines on real-world temporal shifts (e.g., Yearbook, EVIS, FMoW-Time), where it outperforms alternatives, particularly in handling small batch sizes and label shifts. Additionally, it shows versatility by delivering performance gains beyond temporal shifts, including on reproduction datasets, synthetic corruptions, and domain adaptation benchmarks.

**Audience:**

Yes

**Broader Impact Concerns:**

None.

**Claims And Evidence:**

Yes

**Requested Changes:**

1. Elaborate on the reasons behind the limited runtime increase despite nested loops in E-Step and M-Step. For example, explicitly discuss implementation optimizations (e.g., parallelization strategies, GPU acceleration) and their impact.

2. Clarify the definition of "optional" for Steps 17-18 in the algorithm. For example, explain the conditions under which these steps should be included or omitted.

3. Expand on why SAR fails in batch size = 1 scenarios for EVIS and FMoW, beyond the current analysis. Discuss whether hyperparameter tuning or modified mechanisms could improve its performance, and confirm that results are not under-reported (e.g., by verifying experimental setup consistency). This would enhance the robustness of the comparative analysis.

**Strengths And Weaknesses:**

Strengths:

1. **Novel Focus on Temporal Distribution Shifts**: The submission addresses a critical gap in test-time adaptation (TTA) research by focusing on temporal distribution shifts—gradual, time-evolving shifts common in real-world scenarios but underexplored in existing literature. It formalizes the Temporal Test-Time Adaptation (TempTTA) setting, highlighting its unique challenges for established TTA methods.

2. **Innovative Method Design**: STAD (State-space Test-time Adaptation) introduces a probabilistic state-space model (SSM) to track time-varying class prototypes in the representation space, enabling dynamic adaptation of the model’s final layer. This explicit modeling of temporal dynamics, validated via ablation studies, distinguishes it from prior TTA methods that lack such mechanisms.

3. **Strong Empirical Performance**: STAD demonstrates consistent superiority over baselines on real-world temporal shift datasets (Yearbook, EVIS, FMoW-Time), particularly in challenging scenarios like small batch sizes and label shifts. It outperforms both feature extractor-adapting methods (e.g., BN, TENT) and classifier-adapting methods (e.g., LAME, T3A) in these settings.


Weaknesses:

1. The E-Step involves three nested for-loops (iterating over time steps in the sliding window, classes, and samples), and the M-Step also contains for-loops. However, according to the experimental analysis of algorithm runtime, the time consumption does not increase significantly compared to the source model. Why is this the case? Please explain further. Were there any optimizations made in this regard that were not mentioned in the article?

2. I notice that Steps 17-18 in the Algorithm table are marked as "optional". What does "optional" mean, and why can these steps be optional? Please explain further.

3. SAR is specifically designed for the scenario where the batch size is 1. However, judging from the results of EVIS and FMoW in Table 10, this algorithm does not work effectively. Is there a case of under-reporting? Please further explain the reasons.

---

> ### Author Response · Authors · 2025-08-12
>
> We thank you for your thoughtful comments. We address your points below.
>
> > The E-Step involves three nested for-loops [...]
> >
>
> Two of the three for-loops (over classes $K$ and samples $N_t$) are computed in parallel in practice. The loops are written in non-parallel form in Algorithm 1 solely for methodological clarity—if executed sequentially, the runtime would indeed be larger. We understand that this could cause confusion and have updated the algorithms to include only loops that are computed sequentially.
>
> We discuss this parallelization in Appendix D.7, where we reference the specific lines in Algorithm 1 (original submission) that can be parallelized:
>
> *The complexity of STAD-vMF scales linearly with the sliding window size $s$, batch size $N_t$, representation dimension $D$, and number of classes $K$, i.e., $O(s \times N_t \times D \times K)$. The sliding window size $s$ is fixed at $s = 3$, and operations over the batch size, number of classes, and representation dimension can be fully parallelized (lines 7–12 in Algorithm 1).*
>
> In addition, our method updates only the last layer, whereas other TTA methods require a computationally expensive backward pass through the model (see Section 3.1).
>
> > Were there any optimizations made in this regard that were not mentioned in the article?
> >
>
> The parallelization over the number of classes and batch size, as described in Appendix D.7, together with omitting  backpropagation (Section 3.1), enables relatively fast inference.
>
> > I notice that Steps 17–18 in the Algorithm table are marked as "optional". [...] Please explain further.
> >
>
> We explain the optional learning of noise parameters $\kappa^{\text{ems}}$ and $\kappa^{\text{trans}}$ in Appendix C.4:
>
> *We follow suggestions by Gopal & Yang (2014) to keep noise concentration parameters fixed instead of learning them via maximum likelihood (see lines 17 and 18 in Algorithm 1). Keeping them fixed acts as a regularization term, as it controls the size of the cluster (via $\kappa^{\text{ems}}$) and the movement of the prototypes (via $\kappa^{\text{trans}}$).*
>
> Gopal & Yang (2014) argue (Appendix 4) that maximum likelihood could in general be increased by setting $\kappa^{\text{trans}} \to 0$, which corresponds to high transition noise and allows cluster means to differ substantially across time steps. They therefore recommend fixing $\kappa^{\text{trans}}$ to regularize cluster movement, and regularizing cluster size via a prior on $\kappa^{\text{ems}}$. In practice, we found that fixing these parameters (instead of learning them via maximum likelihood) improves stability during adaptation. We therefore denote learning $\kappa^{\text{trans}}$ and $\kappa^{\text{ems}}$ as optional and, by default, only learn $\boldsymbol{\pi}_t$, $\mathbf{W}_t$, and $\mathbf{C}_t$.
>
> > Explain the conditions under which these steps should be included or omitted.
> >
>
> We generally advise fixing the concentration parameters and omitting their optimization via maximum likelihood for the reasons outlined above.
>
> > SAR is specifically designed for the scenario where the batch size is 1. [...] Is there a case of under-reporting?
> >
>
> In short, no—there is no case of underreporting. SAR is only effective for batch size 1 on models with group normalization (GN) or layer normalization (LN), whereas we use pre-trained models from [2] with batch normalization (BN) layers. **Our results are consistent with those reported in the SAR paper [1].**
>
> (i) [1] itself concludes that BN layers are ineffective for small batch sizes: they find that for BN layers, *“the quality of estimated statistics relies on the batch size, and it is hard to use very few samples (i.e., small batch size) to estimate it accurately”* (Section 4). Their Figure 3 shows that entropy minimization on a ResNet-50 with BN layers yields accuracy near zero, while GN and LN remain invariant to batch size.
>
> (ii) [1] reports batch size 1 results for SAR only on GN and LN models (Section 5). No results for SAR on BN models are included.
>
> Since the TTA principle prohibits adapting the source architecture, we follow prior work [3] and use SAR on pre-trained BN models. **We have made this more clear and added the above explanation to the SAR implementation details in Appendix C.4.**
>
> To illustrate this, we train ViT-base (LN) on FMoW and compare it to DenseNet121 (BN), as reported in Table 10. The results show that SAR adapts effectively on LN layers but fails on BN layers.
>
> |  | ViT-base (LN) | DenseNet121 (BN) |
> | --- | --- | --- |
> | Source | $56.87 \pm 0.37$ | $68.94 \pm 0.20$ |
> | SAR | $55.58 \pm 1.26$ | $3.46 \pm 0.03$ |
> | STAD | $56.67 \pm 0.65$ | $68.88 \pm 0.29$ |
>
> [1] Niu, Shuaicheng, et al. "Towards stable test-time adaptation in dynamic wild world." ICLR 2023
>
> [2] Yao, Huaxiu, et al. "Wild-time: A benchmark of in-the-wild distribution shift over time." NeurIPS 2022
>
> [3] Zhao, Hao, et al. "On pitfalls of test-time adaptation." ICML 2023

---

### Review · Reviewer_BpXb · 2025-08-04

**Summary Of Contributions:**

This paper tackles test-time adaptation (TTA) under non-stationary data distributions, which is a realistic setup yet overlooked by the previous TTA literature.  Inspired probabilistic state-space models, this paper proposes State-space Test-time Adaptation (STAD) that models dynamics of temporal distribution shifts in a latent feature space.  With an EM formulation, the class clustering and data distribution are updated in a fully unsupervised manner.  The experiments evaluate the proposed method over three image classification dataset with temporal distributional drift, where the model demonstrates more consistent performance gain over other baselines.  The proposed method also works beyond temporal TTA, showing strong results on two reproduction datasets and a synthetic dataset.

**Audience:**

Yes

**Claims And Evidence:**

Yes

**Requested Changes:**

1. I'd request for clarification on Weakness 1 and 3
2. I'd request for argument against Weakness 2

**Strengths And Weaknesses:**

Strengths:

1. This paper discusses an interesting but often overlooked problem of test-time adaptation, where both sample and label distributions vary over time.  The setup echoes with real-world use case, such change in the visual appearance of landscapes in different seasons.
2. The idea of adapting class assignments and class feature prototypes with EM optimization is technically sound.  Such a formulation is fully unsupervised, without the need for access to class labels during inference / adaptation.
3. This paper presents visualization results well, helping readers understand its high-level idea and appreciating its efficacy.  For example, Figure 1 shows the temporal distributional drift and Figure 8 demonstrate the effective update of class-wise clustering.
4. The experiments demonstrate consistent improvement over different datasets and setups, including datasets for temporal TTA and typical TTA.

---

Weaknesses:

1. The paper writing is confusing:
    - This paper lacks high-level idea, or more precisely, what elements are updated in the E-step and M-step during optimization.  Although the equations provide some hints, it's be more readable if the authors write down them more explicitly
    - Some notations are incorrect: $p(W_{1:T} C_{1:T} |H_{1:T} )$ around eqn 4 and 7 should be $p(W_{1:T}, C_{1:T} |H_{1:T} )$ instead.
2. Although the proposed method is technically sound, its temporal performance is highly-correlated with the source model (see Figure 3).  In other words, all existing methods including STAD can only bring minor performance gain to the source model.  If the source model fails to deal with the current distribution, no TTA method is able to rescue.  These results may hurt the applicability of all these TTA methods in real-world use case.
3. I'd encourage the authors to include necessary background of Kalman Filter in the Appendix.  The connection to Kalman filter described in this paper is still confusing.

---

> ### Author Response · Authors · 2025-08-12
>
> We thank you for your reviewing efforts, they are much appreciated. We address your concerns below.
>
> > *The paper writing is confusing: This paper lacks high-level idea, or more precisely, what elements are updated in the E-step and M-step during optimization. Although the equations provide some hints, it's be more readable if the authors write down them more explicitly*
>
> Here we assume you are asking for low-level details on the E-step and M-step. If we have misunderstood your point, please kindly let us know. In the original version, all update equations for the E-step and M-step of STAD-vMF are listed in Appendix B.2.
>
> **During the rebuttal, we have:**
>
> (i) moved the E-step for STAD-vMF to the main paper (Section 3.3, Equations 8 and 9),
> (ii) added more details on STAD-Gauss in Appendix B.1, including all update equations of the EM algorithm, and
> (iii) modified Algorithm 1 so that it now provides an overview of STAD, and Algorithms 2 and 3 explicitly summarize the EM algorithm for STAD-Gauss and STAD-vMF, respectively.
>
> If you still feel there is something we should elaborate on further when it comes to the EM algorithm, please let us know and we will be happy to provide additional details.
>
> >*Some notations are incorrect: around eqn 4 and 7 $p(\mathbf{W}\_{1:T} \mathbf{C}\_{1:T} \mid \mathbf{H}\_{1:T})$ should be $p(\mathbf{W}\_{1:T}, \mathbf{C}\_{1:T} \mid \mathbf{H}\_{1:T})$ instead.*
>
> Thanks for pointing out the comma typo. We fixed it.
>
> > *Although the proposed method is technically sound, its temporal performance is highly-correlated with the source model (see Figure 3). In other words, all existing methods including STAD can only bring minor performance gain to the source model. If the source model fails to deal with the current distribution, no TTA method is able to rescue. These results may hurt the applicability of all these TTA methods in real-world use case.*
>
> We agree with the reviewer that the performance of any TTA method is inherently bounded by the quality of the source model. However, **this is not a limitation specific to our method, but applies to all TTA approaches**: since TTA operates in a fully unsupervised setting at test time, it can only leverage the information contained in the source model and the incoming unlabeled data stream. TTA is not designed to “rescue” a completely failing model; instead, its goal is to **mitigate the effects of distribution shift to the extent possible without labeled supervision**.
>
> The TTA field is well established, with over 100 papers published last year \[1] at top-tier conferences. Our results (including Fig. 3) confirm that STAD is competitive with, and often outperforms, prior methods under this shared constraint.
>
> \[1] Xiao, Zehao, and Cees GM Snoek. "Beyond model adaptation at test time: A survey." *arXiv preprint arXiv:2411.03687* (2024).
>
> > *I'd encourage the authors to include necessary background of Kalman Filter in the Appendix.*
> > *The connection to Kalman filter described in this paper is still confusing.*
>
> In the revised manuscript, we have expanded Appendix B.1 to provide additional details, including the full inference step of STAD-Gauss. From the update equations of the EM algorithm detailed in Appendix B.1, one can see that STAD-Gauss leverages the standard Kalman filtering and smoothing equations, with references provided where appropriate (see Eqs. 14–19). The main difference from the traditional Kalman filter is that instead of filtering a single trajectory, we filter \$K\$ trajectories—one for each class \$k\$—and place a mixture distribution over them.
>
> We hope this clarifies the connection between our model and the Kalman filter, and we would be happy to elaborate further if anything remains unclear.
>
> If we have addressed your concerns, we would appreciate it if you could reassess your evaluation.

---

### Decision · Action_Editor_K5Bf · 2025-09-16

**Recommendation:** Accept as is

**Audience:**

Yes

**Audience Explanation:**

All three reviewers agree that the submission is of interest to at least part of the TMLR audience:

- "(TTA) under non-stationary data distributions [...] is a realistic setup yet overlooked by the previous TTA literature." (BpXb)
- "The submission addresses a critical gap in test-time adaptation (TTA) research by focusing on temporal distribution shifts—gradual, time-evolving shifts common in real-world scenarios but underexplored in existing literature." (q6BP)
- "[The] problem this paper studies is important, as real-world distribution shifts are not static but could vary over time." (F8Aa)

Reviewer BpXB's concern over the applicability of the proposed TTA approach in real-world use cases has been addressed to their satisfaction by the authors.

**Claims And Evidence:**

Yes

**Claims Explanation:**

All three reviewers agree that the submission meets the bar in terms of claims and evidence:

- "This paper presents visualization results well, helping readers understand its high-level idea and appreciating its efficacy." (BpXb)
- "The experiments demonstrate consistent improvement over different datasets and setups, including datasets for temporal TTA and typical TTA." (BpXb)
- "STAD demonstrates consistent superiority over baselines on real-world temporal shift datasets (Yearbook, EVIS, FMoW-Time), particularly in challenging scenarios like small batch sizes and label shifts." (q6BP)
- "This paper is well-written, with detailed approach description and experimental evaluation." (F8Aa)

Reviewer questions and concerns regarding the E and M steps, the time complexity of the proposed approach, and the framing of the approach using state-space models were addressed to their satisfaction by the author's response.